# Perturbed Gradient Descent via Convex Quadratic Approximation for Nonconvex Bilevel Optimization

**Nazanin Abolfazli**[*]                                              *nazaninabolfazli@arizona.edu*
*Department of Systems and Industrial Engineering, The University of Arizona*
**Sina Sharifi**[*]                                                            *sshari12@jhu.edu*
*Department of Electrical and Computer Engineering at Johns Hopkins University*
**Mahyar Fazlyab**                                                          *mahyarfazlyab@jhu.edu*
*Department of Electrical and Computer Engineering at Johns Hopkins University*
**Erfan Yazdandoost Hamedani**                                          *erfany@arizona.edu*
*Department of Systems and Industrial Engineering, The University of Arizona*

**Reviewed on OpenReview:** *https: // openreview. net/ forum? id= sFtPtOHzYO*

## Abstract

Bilevel optimization is a fundamental tool in hierarchical decision-making and has been widely applied to machine learning tasks such as hyperparameter tuning, meta-learning, and adversarial learning. Although significant progress has been made in bilevel optimization, existing methods predominantly focus on the nonconvex-strongly convex, or the nonconvex-PL settings; the more general nonconvex-nonconvex framework is underexplored. In this paper, we address this gap by developing an efficient gradient-based method to decrease the upper-level objective, coupled with a convex Quadratic Program (QP) that minimally perturbs the gradient descent directions to reduce the suboptimality of the condition imposed by the lower-level problem. We provide a rigorous convergence analysis, demonstrating that under the existence of a KKT point and a regularity assumption (norm-squared gradient of the lower-level satisfies PL), our method achieves an iteration complexity of $\mathcal{O}(1/\epsilon^{1.5})$ in terms of the squared norm of the KKT residual for the reformulated problem. Moreover, even in the absence of the regularity assumption, we establish an iteration complexity of $\mathcal{O}(1/\epsilon^3)$ for the same metric. Through extensive numerical experiments on convex and nonconvex synthetic benchmarks and data hyper-cleaning tasks, we illustrate the efficiency and scalability of our approach.

## 1 Introduction

Bilevel optimization is a fundamental framework in hierarchical decision-making, in which one optimization problem is nested within another. This class of problem finds a plethora of applications in engineering (Pandžić et al., 2018), economics (Von Stackelberg & Von, 1952), transportation (Sharma et al., 2015), and machine learning (Finn et al., 2017; Rajeswaran et al., 2019; Fei-Fei et al., 2006; Hong et al., 2020; Bengio, 2000; Hao et al., 2024; Zhang et al., 2022). A broad category of bilevel optimization problems can be written as optimization problems of the following form, also known as optimistic bilevel optimization.

$$\min_{x \in \mathbb{R}^n, y \in \mathcal{Y}^\star(x)} f(x,y) \quad \text{s.t.} \quad \mathcal{Y}^\star(x) = \arg\min_{y \in \mathbb{R}^m} g(x,y), \tag{BLO}$$

where $f, g : \mathbb{R}^n \times \mathbb{R}^m \to \mathbb{R}$ represent the upper-level and lower-level objective functions. The implicit objective approach (Dempe, 2002) reformulated BLO by minimizing the problem defined below

$$\min_{x \in \mathbb{R}^n} \ell(x) \quad \text{where} \quad \ell(x) := \min_{y \in \mathcal{Y}^\star(x)} f(x,y),$$

---

[*]Equal contribution.

where $\ell : \mathbb{R}^n \to \mathbb{R}$ denotes the implicit objective function. Bilevel optimization problems are inherently nonconvex and computationally challenging, largely due to the complex interdependence between the upper-level and lower-level problems.

The majority of recent works in bilevel optimization concentrate on scenarios where the lower-level problem is strongly convex. Under the strong convexity of $g(x, \cdot)$, the solution of the lower-level problem $\mathcal{Y}^\star(x)$ is a singleton, and bilevel optimization reduces to minimizing $\ell(x)$ whose gradient can be calculated with implicit gradient method (Pedregosa, 2016; Gould et al., 2016; Ghadimi & Wang, 2018; Lorraine et al., 2020; Ji et al., 2021; Li et al., 2022; Abolfazli et al., 2023). Recently, to relax the strong convexity assumption in the lower-level problem, some works focused on bilevel optimization with merely convex lower-level objectives. However, this introduces significant challenges, particularly the presence of multiple lower-level local optima (i.e., a non-singleton solution set) may hinder the adoption of implicit-based approaches that rely on implicit function theorem (Sow et al., 2022; Liu et al., 2023; Lu & Mei, 2024). Bilevel optimization problems with nonconvex lower-level objectives are common in various machine learning applications, such as hyperparameter optimization in deep neural network training (Vicol et al., 2022), continual learning (Borsos et al., 2020; Hao et al., 2024), and more. However, the above bilevel optimization methods primarily rely on the assumption of lower-level strong convexity or convexity, which significantly limits their effectiveness in handling nonconvex lower-level problems. Recently, some works studied bilevel optimization with nonconvex lower-level problems (Liu et al., 2022; Chen et al., 2023; Huang, 2023; 2024). Next, we provide the existing approaches for solving the bilevel optimization problem and recent works closely related to ours.

## 1.1 Existing Approaches for Solving Bilevel Optimization Problems

In this subsection, we briefly discuss traditional approaches for solving bilevel optimization problems.

**Hyper-gradient Descent:** Assume that the minimum of $g(x, \cdot)$ is unique for all $x$, resulting in the optimal solution map $\mathcal{Y}^\star(x)$ of the lower-level problem being single-valued and continuously differentiable. The most straightforward approach to solving BLO is to perform a gradient descent on the implicit objective function $\ell(x)$.

$$\nabla \ell(x) = \nabla_x f(x, y^\star(x)) + D_x y^\star(x) \nabla_y f(x, y^\star(x)).$$

If the Hessian matrix $\nabla^2_{yy} g(x, y^\star(x))$ is invertible, the implicit function theorem (Rudin, 1976) guarantees that the map $x \mapsto y^\star(x)$ is continuously differentiable. Moreover, the Jacobian of the solution map, $D_x y^\star(x)$, can be derived by differentiating the implicit equation with respect to $x$.

$$\nabla^2_{yx} g(x, y^\star(x)) + D_x y^\star(x) \nabla^2_{yy} g(x, y^\star(x)) = 0.$$

This approach is sometimes known as the hyper-gradient descent. However, hyper-gradient descent is computationally expensive due to the need to compute $D_x y^*(x)$ at each step. To alleviate the computational burden, various approximation methods have been developed to avoid direct inversion of the Hessian (Pedregosa, 2016; Rajeswaran et al., 2019; Grazzi et al., 2020; Ghadimi & Wang, 2018; Lorraine et al., 2020). In addition, the exact computation of $y^\star(x)$ can be mitigated by considering a common surrogate map through replacing the optimal lower-level solution with an approximated solution $y$,

$$F(x, y) := \nabla_x f(x, y) - \nabla^2_{yx} g(x, y)^\top v(x, y), \tag{1a}$$

$$\nabla^2_{yy} g(x, y) v(x, y) = \nabla_y f(x, y). \tag{1b}$$

Under suitable assumptions on the upper-level objective function $f$, this estimate can be controlled by the distance between the optimal solution $y^\star(x)$ and the approximated solution $y$. In equation 1 the effect of Hessian inversion is presented in a separate term $v(x, y)$ which solves a parametric quadratic problem and can be approximated using one or multiple steps of gradient descent (Abolfazli et al., 2023; Li et al., 2022; Arbel & Mairal, 2021).

**Value Function Approach:** Another approach is based on the observation that BLO is equivalent to the following constrained optimization:

$$\min_{x,y} \ f(x, y) \quad \text{s.t.} \quad g(x, y) - g^*(x) \leq 0, \tag{2}$$

Table 1: Comparison of bilevel algorithms with nonconvex lower-level, specifically without the presence of lower-level constraints. **PL** and **SC** stands for the Polyak-Łojasiewicz and strong convexity, respectively. $\sigma(A)$ represents the singular values of matrix A. Complexity is based on finding an $\epsilon$-stationary solution such that $\|\nabla\ell(x)\|^2 \leq \epsilon$ or its equivalent variants. The notation $\tilde{\mathcal{O}}$ omits the dependency on $\log(\varepsilon^{-1})$. For more discussion on related works see Section 1.2.1.

| Algorithm | $g(x,\cdot)$ | Additional Assumptions | Oracle | Loop(s) | Complexity |
|---|---|---|---|---|---|
| BOME (Liu et al., 2022) | PL | Bounded $|f|$ and $|g|$, lower-level unique solution | 1st | Double | $\mathcal{O}(\epsilon^{-3})$ |
| V-PBGD (Shen & Chen, 2023) | PL | See Remark 1.1 | 1st | Double | $\tilde{\mathcal{O}}(\epsilon^{-3/2})$ |
| $F^2$BA (Chen et al., 2024) | PL | $\mu$-PL penalty function, $f$ has Lipschitz Hessians in y | 1st | Double | $\tilde{\mathcal{O}}(\epsilon^{-1})$ |
| GALET (Xiao et al., 2023a) | PL + SC | $\inf_{x,y}\{\sigma^+_{\min}(\nabla^2_{yy}g(x,y))\} > 0 \quad \forall x,y$ | 2nd | Triple | $\tilde{\mathcal{O}}(\epsilon^{-1})$ |
| HJFBiO (Huang, 2024) | PL | $\sigma(\nabla^2_{yy}g(x,y^\star(x))) \in [\mu, L_g]$ | 1st | Single | $\mathcal{O}(\epsilon^{-1})$ |
| **Ours-Theorem4.2** | nonconvex | Regularity Condition, see Remark 4.4 | 2nd | Single | $\mathcal{O}(\epsilon^{-3/2})$ |
| **Ours-Theorem 4.5** | nonconvex | - | 2nd | Single | $\mathcal{O}(\epsilon^{-3})$ |

where $g^*(x) = \min_y g(x,y)$. Compared to hyper-gradient approaches, this method does not require computing the implicit derivative $D_x y^\star(x)$. As shown in (Ye & Zhu, 1995), one cannot establish a KKT condition of BLO through this reformulation since none of the standard constraint qualifications (Slater's, LICQ, MFCQ, CRCQ) hold for this reformulation. Moreover, Xiao et al. (2023a) showed that the calmness condition, which is the weakest constraint qualification, does not hold for bilevel optimization with a nonconvex lower-level when employing the value function-based reformulation.

**Stationary-Seeking Methods:** An alternative method is to replace the lower-level problem in BLO with the stationarity condition. As a result, the bilevel optimization problem can be reformulated as the following constrained, nonconvex single-level optimization problem:

$$\min_{x,y} \ f(x,y) \quad \text{s.t.} \quad \nabla_y g(x,y) = 0. \tag{3}$$

The reformulated problem equation 3 coincides with the original bilevel optimization problem BLO under the assumption that $g(x,\cdot)$ is convex and/or satisfies the weak PL condition (Csiba & Richtárik, 2017) for any $x$. In scenarios where $g(x,\cdot)$ lacks these conditions, the reformulation in equation 3 corresponds to finding a minimizer of the upper-level function over stationary solutions of the lower-level problem.

## 1.2 Related Works

In this section, we present a comprehensive review of recent works that are closely related to bilevel optimization problems with a nonconvex lower-level. While most works assume a unique lower-level solution, newer studies address cases where this assumption does not hold (Sow et al., 2022; Chen et al., 2023; Shen et al., 2024; Liu et al., 2022; Xiao et al., 2023a). Some works focus on bilevel optimization with a convex lower-level problem, which introduces the challenge of multiple lower-level solutions (Liu et al., 2020; Sow et al., 2022; Liu et al., 2023; Shen et al., 2024; Shen & Chen, 2023; Chen et al., 2023; Lu & Mei, 2024). However, Chen et al. (2023) has shown that additional assumptions on the lower-level problem are necessary to ensure meaningful guarantees.

Beyond lower-level convexity, recently, several studies have studied bilevel optimization with a nonconvex lower-level problem satisfying the PL condition. More specifically, Liu et al. (2022) proposed a first-order method and established the first non-asymptotic convergence guarantee of $\mathcal{O}(\epsilon^{-3})$ for bilevel optimization satisfying the PL condition. They further assume that the lower-level solution is unique, and both the upper and lower-level objective functions are bounded. Later, Shen & Chen (2023) introduced a penalty-based algorithm in which the lower-level objective $g$ satisfies the PL condition, where the method relies solely on first-order oracles with an iteration complexity of $\mathcal{O}(\epsilon^{-3/2})$ in terms of $\epsilon$-stationary of the penalized objective function. Kwon et al. (2023) studied the nonconvex bilevel optimization with nonconvex lower-level satisfying proximal error-bound (EB) condition that is analogous to PL condition when the lower-level is unconstrained. Their approach guarantees convergence to an $\epsilon$-stationary point of the penalty function, requiring $\tilde{\mathcal{O}}(\epsilon^{-3/2})$ first-order gradient oracle calls. Further Chen et al. (2024) showed that the proximal operator in (Kwon et al., 2023) is unnecessary, and their method can converge under the PL condition with an improved complexity of $\tilde{\mathcal{O}}(\epsilon^{-1})$ in terms of $\epsilon$-implicit gradient, i.e., $\|\nabla\ell(x)\|^2 \leq \epsilon$. However, they assume that the penalty function

$h_\theta = \theta f + g$ satisfies the $\mu$–PL condition, which is considerably stronger and more restrictive than simply assuming that the lower-level function $g$ is $\mu$–PL. This is because it is not even straightforward to ensure that the sum of two PL functions remains PL. Moreover, assuming that the Hessian of the upper-level objective function is Lipschitz continuous is a particularly restrictive condition that further narrows the class of problems to which the analysis applies. Xiao et al. (2023a) proposed a generalized alternating method and obtained an $\epsilon$-stationary point within $\tilde{\mathcal{O}}(\epsilon^{-1})$ in terms of $\epsilon$-KKT of equation 3 under PL condition of the lower-level problem. Huang (2023) introduced a class of momentum-based gradient methods for nonconvex bilevel optimization problems, where both upper-level and lower-level problems are nonconvex, and the lower-level problem satisfies the PL condition. Furthermore, they assume that $\nabla_{yy}^2 g(x, y^*(x))$ is non-singular at the minimizer of $g$. Their method achieves a complexity of $\tilde{\mathcal{O}}(\epsilon^{-1})$ in finding an $\epsilon$-stationary of the implicit gradient. However, their proposed method requires computing expensive projected Hessian and Jacobian matrices along with their inverses. Moreover, computing the SVD decomposition of the Hessian matrix at each step imposes a $\mathcal{O}(d^3)$ complexity where $d = \max\{m, n\}$. More recently, Huang (2024) claimed that the projection operator can remove expensive SVD decomposition in (Huang, 2023) and proposed a Hessian/Jacobian-free bilevel method achieving a complexity of $\mathcal{O}(\epsilon^{-1})$ in finding $\epsilon$-stationary solution under the same setting. A concise comparison between our proposed method and related works is summarized in Table 1. Some techniques additionally handle equality and inequality constraints at the lower-level (Xiao et al., 2023b; Khanduri et al., 2023; Xu & Zhu, 2023; Kornowski et al., 2024).

### 1.2.1 Discussion on Related Works

**Remark 1.1.** *We make the following remarks regarding Table 1:*

- *BOME (Liu et al., 2022): They assume a unique lower-level solution and bounded upper- and lower-level objective functions. BOME is implemented as a double-loop algorithms, which approximate the lower-level optimal value function. In these approaches, each outer iteration incurs a computational cost on the order of $m \times T$, where $m$ is the lower-level problem dimension and $T$ is the number of inner iterations required to obtain an approximate solution. In contrast, our method adopts a single-loop design. Although it involves second-order derivatives in theory, modern automatic differentiation frameworks (e.g., PyTorch (Paszke et al., 2019) ) allow direct computation of the required matrix–vector products, eliminating the need to explicitly form or store second-order tensors. This substantially reduces computational overhead.*

- *V-PBGD (Shen & Chen, 2023): They establish conditions under which global or local minimizers of the penalized problem correspond to global or local minimizers of the original bilevel problem. However, the relation between the stationary points of the penalized problem with those of the original bilevel problem remains unsettled. In their setting, a stationary point merely indicates that the gradient of the penalized objective vanishes, but this does not ensure that the penalty function $p(x, y) := g(x, y) - g(x)$ is zero. Consequently, such a stationary point does not satisfy the essential lower-level constraint $y \in \arg\min g(x, \cdot)$ and therefore is not a valid solution to the original bilevel problem.*

- *GALET (Xiao et al., 2023a): In their work, the assumption $\inf_{(x,y)}\{\sigma_{\min}^+(\nabla_{yy}^2 g(x,y))\} > 0$ is a strong global condition asserting that the Hessian of $g$ w.r.t $y$ is uniformly nondegenerate. In other words, for every fixed $x$, the function $g(x, \cdot)$ exhibits a form of strong convexity which guarantees that the lower-level problem $\min_y g(x, y)$ has a unique minimizer $y^*(x)$ that depends smoothly on $x$. Thus, this assumption, along with PL condition, and Lipschitz continuity of the Hessian implies that the GALET assumes the strongly convex setting. For more details on this, see Appendix B of (Huang, 2024).*

- *HJFBiO (Huang, 2024): For $\mu$-PL function $g$, they assumed that all singular values of $\nabla_{yy}^2 g(x, y^*(x))$ lie in $[\mu, L_g]$ implying the Hessian is positive definite at the minimizer $y^*(x)$, ensuring local strong convexity of $g(x, y)$ around $y^*(x)$. This is stronger than the PL inequality (which only ensures gradient growth) but weaker than strong convexity (which requires positive definiteness everywhere). The assumption guarantees that $y^*(x)$ is an isolated local minimizer, enabling smooth dependence of $y^*(x)$ on $x$.*

### 1.3 Contribution

In this paper, we study a general bilevel optimization problem with smooth objective functions. While existing approaches largely focus on the nonconvex-PL setting, we consider a more general nonconvex-nonconvex bilevel

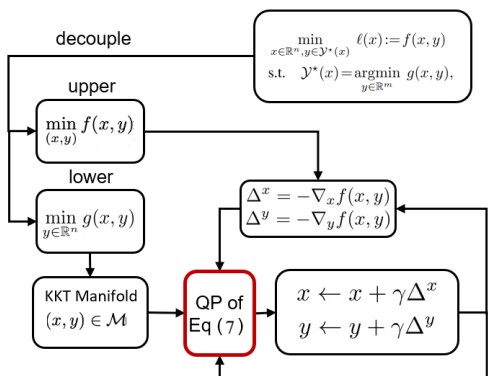

Figure 1: Overview of the proposed method. The QP takes the gradient directions, perturbs them according to the lower-level problem, and then the variables are updated using the new directions.

framework. We address this gap by developing an efficient gradient-based framework that directly minimizes the upper-level objective while respecting the bilevel structure. The method introduces a convex Quadratic Program (QP) subproblem at each iteration with a closed-form solution that minimally perturbs the gradient descent direction to correct for suboptimality arising from the lower-level problem's constraints. Our approach builds upon the principles of Relaxed Gradient Flow (RXGF) (Sharifi et al., 2025), but departs from it in key ways: instead of relying on continuous-time dynamics and ODE solvers, we design a discrete-time algorithm that is computationally efficient and scalable to high-dimensional problems. Furthermore, we introduce a modified dynamic constraint–barrier mechanism that compensates for discretization-induced accumulation errors, ensuring improved numerical stability and convergence guarantee under weaker assumptions. We establish that, under the existence of a KKT point and a regularity assumption, our method achieves a convergence rate of $\mathcal{O}(1/\epsilon^{1.5})$ in terms of the squared norm of the KKT residual. Moreover, even without the regularity assumption, we demonstrate that our method attains a convergence rate of $\mathcal{O}(1/\epsilon^3)$ for the same metric. Fig. 1 presents an overview of the methods.

## 2 Preliminaries

### 2.1 Assumptions and Definitions

This subsection outlines the definitions and assumptions required throughout the paper.

**Assumption 2.1** (Upper-level Objective). *$f : \mathbb{R}^n \times \mathbb{R}^m \to \mathbb{R}$ is a continuously differentiable function such that $\bar{f} \triangleq \inf_{(x,y)} f(x,y) > -\infty$. $\nabla_x f(\cdot, \cdot)$ and $\nabla_y f(\cdot, \cdot)$ are $L_x^f$ and $L_y^f$-Lipschitz continuous, respectively. Moreover, there exists $C_f > 0$ such that $\|\nabla f(x,y)\| \leq C_f$ for any $(x,y)$.*

**Assumption 2.2** (Lower-level Objective). *For any $x$, the function $g(x, \cdot)$ is twice continuously differentiable. $\nabla_y g(x, \cdot)$, and $\nabla_y g(\cdot, y)$, are $L_{yy}^g$- and $L_{yx}^g$-Lipschitz continuous for all $(x,y)$, respectively. There exists $C_g > 0$ such that $\|\nabla_y g(x,y)\| \leq C_g$ for all $(x,y)$.*

**Assumption 2.3.** *There exists $(\bar{x}, \bar{y})$ and $\bar{\nu} \in \mathbb{R}^m$ such that $\nabla_y g(\bar{x}, \bar{y}) = 0$, $\nabla_x f(\bar{x}, \bar{y}) + \nabla_{yx}^2 g(\bar{x}, \bar{y})^\top \bar{\nu} = 0$, and $\nabla_y f(\bar{x}, \bar{y}) + \nabla_{yy}^2 g(\bar{x}, \bar{y})^\top \bar{\nu} = 0$.*

Our goal is to find an $\epsilon$-approximate KKT point of problem 3, which we formally define as follows

**Definition 2.4.** *$(x_\epsilon, y_\epsilon, \nu_\epsilon) \in \mathbb{R}^n \times \mathbb{R}^m \times \mathbb{R}^m$ is called an $\epsilon$-KKT point of 3 if $\|\nabla_y g(x_\epsilon, y_\epsilon)\|^2 \leq \epsilon$, $\|\nabla_x f(x_\epsilon, y_\epsilon) + \nabla_{yx}^2 g(x_\epsilon, y_\epsilon)^\top \nu_\epsilon\|^2 \leq \epsilon$, and $\|\nabla_y f(x_\epsilon, y_\epsilon) + \nabla_{yy}^2 g(x_\epsilon, y_\epsilon)^\top \nu_\epsilon\|^2 \leq \epsilon$.*

The sufficient conditions for the existence of a KKT solution for problem 3 have been studied in several studies (Ye & Zhu, 1995; Xiao et al., 2023a). For example, in (Liu et al., 2022), a constant rank constraint qualification (CRCQ) is assumed to guarantee the existence of KKT points. Meanwhile, Xiao et al. (2023a) explored the calmness condition and demonstrated that the PL condition for $g(x, \cdot)$ ensures the existence of

a KKT point. However, since our focus is on finding an $\epsilon$-KKT point as defined above, we adopt a more general assumption and only require that such a point exists, as stated in Assumption 2.3.

## 2.2 Safe Gradient Flow for Bilevel Optimization

In the context of bilevel optimization, Sharifi et al. (2025) has recently introduced an ODE for solving the following single-level reduction of the bilevel problem in BLO:

$$\min_{(x,y)} f(x,y) \quad \text{s.t.} \quad h(x,y) := \|\nabla_y g(x,y)\|_2^2 = 0. \tag{4}$$

Specifically, this ODE is obtained as the solution to the following convex QP:

$$(\dot{x}, \dot{y}) := \operatorname*{arg\,min}_{(\dot{x}_d, \dot{y}_d)} \frac{1}{2}\|\dot{x}_d + \nabla_x f\|_2^2 + \frac{1}{2}\|\dot{y}_d + \nabla_y f\|_2^2 \tag{5}$$
$$\text{s.t.} \ \nabla_x h^\top \dot{x}_d + \nabla_y h^\top \dot{y}_d + \alpha h = 0.$$

The quadratic program seeks velocity vectors $(\dot{x}, \dot{y})$ that stay as close as possible to the gradient descent direction $(-\nabla_x f, -\nabla_y f)$ while enforcing the constraint $\dot{h} + \alpha h = 0$. This condition yields $h(t) = h(0)e^{-\alpha t}$, ensuring exponential decay of $h$ to zero and, consequently, satisfaction of the lower-level constraints. The closed-form solution of the QP leads to the ODE,

$$\dot{x} = -\nabla_x f - \lambda \nabla_x h, \quad \dot{y} = -\nabla_y f - \lambda \nabla_y h,$$
$$\lambda = \begin{cases} \frac{-\nabla_x h^\top \nabla_x f - \nabla_y h^\top \nabla_y f + \alpha h}{\|\nabla_x h\|^2 + \|\nabla_y h\|^2} & h \neq 0 \\ 0 & h = 0 \end{cases} \tag{6}$$

where $\lambda$ is the optimal Lagrangian multiplier of equation 5. Owing to the constraint in equation 5, this ODE ensures $\dot{h} + \alpha h = 0$ along the trajectories, implying exponential convergence of $h$ to zero. As another appealing feature, equation 6 does not involve any matrix inversion calculation due to using a single constraint in equation 5. However, since the Lagrangian multiplier $\lambda$ becomes unbounded as $h$ approaches zero, Sharifi et al. (2025) relaxes the equality constraint in equation 4 to an inequality constraint $h(x,y) \leq \epsilon$, $\epsilon > 0$. Then, it is proven in Sharifi et al. (2025) that when $g$ is strongly convex, the resulting ODE converges to an $\epsilon$-neighborhood of a stationary solution of BLO at an $\mathcal{O}(1/t)$ rate ($\epsilon > 0$ arbitrary). One might be tempted to discretize equation 6 directly using, for example, the Euler method to derive an iterative algorithm. However, ensuring exponential convergence of $h$ to zero imposes strong assumptions on the lower-level objective $g$. Specifically, the full rank condition of the matrix $[\nabla_{yx}^2 g \ \nabla_{yy}^2 g]$ (which holds if $g(x, \cdot)$ is strongly convex) guarantees that the QP in equation 5 is always feasible, which in turn ensures the exponential convergence of $h$ to zero.

In the following section, we will design an alternative QP directly in discrete time that guarantees convergence of the resulting iterative algorithm to a stationary point *without* requiring strong assumptions on the lower-level objective $g$.

## 3 Proposed Method

Our goal is to design an iterative algorithm of the form

$$x_{k+1} = x_k + \gamma \Delta_k^x, \quad y_{k+1} = y_k + \gamma \Delta_k^y,$$

where $\gamma > 0$ is the stepsize and $\Delta_k^x, \Delta_k^y$ are search directions. Inspired by equation 5, we propose the following QP to obtain these directions,

$$(\Delta_k^x, \Delta_k^y) = \operatorname*{arg\,min}_{\Delta^x, \Delta^y} \frac{1}{2}\|\Delta^x + \nabla_x f_k\|_2^2 + \frac{1}{2}\|\Delta^y + \nabla_y f_k\|_2^2 \tag{7}$$
$$\text{s.t.} \ \nabla_x h_k^\top \Delta^x + \nabla_y h_k^\top \Delta^y + \alpha \rho_k \leq 0,$$

where $\nabla_x f_k = \nabla_x f(x_k, y_k)$, $\nabla_y f_k = \nabla_y f(x_k, y_k)$, $\nabla_x h_k = \nabla_x h(x_k, y_k)$, $\nabla_y h_k = \nabla_y h(x_k, y_k)$, and $\rho_k = \rho(x_k, y_k)$, to be designed, is positive whenever $h_k = h(x_k, y_k) > 0$ and is zero otherwise, ensuring reduction of infeasibility. The primal and dual solution to this QP is

$$\Delta_k^x = -\nabla_x f_k - \lambda_k \nabla_x h_k, \quad \Delta_k^y = -\nabla_y f_k - \lambda_k \nabla_y h_k, \tag{8a}$$

$$\lambda_k = \frac{\left[-\nabla_x h_k^\top \nabla_x f_k - \nabla_y h_k^\top \nabla_y f_k + \alpha \rho_k\right]_+}{\|\nabla_x h_k\|^2 + \|\nabla_y h_k\|^2}, \tag{8b}$$

where we use the notation $[x]_+ \triangleq \max\{0, x\}$. Our proposed method is outlined in Algorithm 1.

**Choice of $\rho_k$:** The key distinction between equation 7 and equation 5 lies in the choice of $\rho_k$, which plays a crucial role in ensuring convergence. Intuitively, $\rho_k$ acts as a feedback term that controls how aggressively the dynamics reduce the violation measure $h(x, y)$. Different choices of $\rho_k$ correspond to different decay laws for $h(x, y)$, analogous to choosing a class-$\mathcal{K}$ function $\alpha(\cdot)$ in a differential inequality $\dot{h}(x, y) + \alpha(h(x, y)) \leq 0$. For example, a linear dependence on $h$ leads to exponential decay, while a square-root dependence leads to finite-time decay in the continuous-time counterpart. Motivated by this perspective, to balance objective function value reduction with infeasibility reduction in equation 4, we consider two choices for $\rho(\cdot)$:

- $\rho(x, y) = \|\nabla h(x, y)\|^2$. As we will establish, this choice of $\rho$ guarantees the reduction of the first-order KKT condition, leading to a stationary point of the constraint, i.e., $\|\nabla h(x, y)\| \leq \epsilon$. Combined with the PL condition on $h$, this can be translated into an infeasibility result.

- $\rho(x, y) = \|\nabla h(x, y)\| \sqrt{h(x_0, y_0)}$. Our second choice removes the requirement of such a regularity assumption and instead requires a warm start, i.e., given $x_0 \in \mathbb{R}^n$ find $y_0$ such that $h(x_0, y_0) = \|\nabla_y g(x_0, y_0)\|^2$ is sufficiently small, which can be computed efficiently by once running (accelerated) gradient descent method for solving the minimization $\min_y g(x_0, y)$.

We note that both choices ensure that the QP equation 7 is always feasible. This follows from the fact that when $\nabla_x h = 0$ and $\nabla_y h = 0$, we also have $\rho = 0$. Algorithm 1 outlines the proposed method, detailing the iterative procedure for solving the bilevel optimization problem BLO.

**Remark 3.1.** *[Computing $\nabla_x h$ and $\nabla_y h$] The proposed methods require computing the gradients of $h$ as*

$$\nabla_x h = 2\nabla_{yx}^2 g^\top \nabla_y g, \quad \nabla_y h = 2\nabla_{yy}^2 g^\top \nabla_y g.$$

*At first glance, this might suggest the need to store the Jacobian of $g$, specifically $\nabla_{yy}^2 g$ and $\nabla_{yx}^2 g$. However, modern automatic differentiation frameworks such as PyTorch (Paszke et al., 2019) enable direct computation of the required matrix-vector products, eliminating the need to explicitly store these second-order derivatives. This significantly reduces the computational burden. More specifically, using PyTorch's `torch.autograd.grad` with `grad_outputs=dgdy`, we can efficiently compute the necessary matrix-vector products without constructing the full Jacobian.*

## 4 Convergence Analysis

In this section, we establish the convergence properties of our proposed algorithm for solving equation 4. Our objective is to find a pair $(\bar{x}, \bar{y})$ that satisfies the $\epsilon$-KKT conditions defined in Definition 2.4. First, we prove a convergence rate of $\mathcal{O}(1/K^{2/3})$ under a regularity assumption when choosing $\rho(x, y) = \|\nabla h(x, y)\|^2$. Furthermore, we show that by setting $\rho(x, y) = \|\nabla h(x, y)\| \sqrt{h(x_0, y_0)}$, we achieve a convergence rate of $\mathcal{O}(1/K^{1/3})$ without requiring any regularity assumption. We first present the following Lemma on Lipschitz continuity of $h$.

---

**Algorithm 1** Bilevel Approximation via Perturbed Gradient Descent

---

1: **Input:** $\gamma, \alpha, C_0 > 0$, $\rho : \mathbb{R}^n \times \mathbb{R}^m \to \mathbb{R}$
2: **Initialization:** $x_0 \in \mathbb{R}^n$, $y_0 \in \mathbb{R}^m$ such that $\|\nabla_y g(x_0, y_0)\|^2 \leq \alpha^2 C_0$
3: **for** $k \geq 0$ **do**
4:     Compute $\lambda_k$ using equation 8b.
5:     $\Delta_k^x \leftarrow -\nabla_x f(x_k, y_k) - \lambda_k \nabla_x h(x_k, y_k)$.
6:     $\Delta_k^y \leftarrow -\nabla_y f(x_k, y_k) - \lambda_k \nabla_y h(x_k, y_k)$.
7:     $(x_{k+1}, y_{k+1}) \leftarrow (x_k, y_k) + \gamma(\Delta_k^x, \Delta_k^y)$.
8: **end for**

---

**Lemma 4.1.** *Suppose Assumption 2.2 holds. Then, the function $h : \mathbb{R}^n \times \mathbb{R}^m \to \mathbb{R}$ defined by $h(x, y) = \|\nabla_y g(x, y)\|^2$ is Lipschitz continuous with constant $L_h = 2C_g(L_{yy}^g + L_{yx}^g)$.*

*Proof.* Consider any two points $(x_1, y_1)$ and $(x_2, y_2)$ in $\mathbb{R}^n \times \mathbb{R}^m$. We have $h(x_1, y_1) - h(x_2, y_2) = \|\nabla_y g(x_1, y_1)\|^2 - \|\nabla_y g(x_2, y_2)\|^2$. This can be rewritten using the identity $a^2 - b^2 = (a + b)(a - b)$ followed by utilizing Assumption 2.2 along with the application of triangle inequality as

$$
\begin{aligned}
|h(x_1, y_1) - h(x_2, y_2)| &\le 2C_g \|\nabla_y g(x_1, y_1) - \nabla_y g(x_2, y_2)\| \\
&= 2C_g \|\nabla_y g(x_1, y_1) - \nabla_y g(x_1, y_2) + \nabla_y g(x_1, y_2) - \nabla_y g(x_2, y_2)\| \\
&\le 2C_g\big(L_{yy}^g \|y_1 - y_2\| + L_{yx}^g \|x_1 - x_2\|\big) \le 2C_g(L_{yy}^g + L_{yx}^g)\|(x_1, y_1) - (x_2, y_2)\|
\end{aligned}
$$

which implies the result. $\qquad\square$

Next, we establish a convergence bound for the magnitude of the update direction, $\|\Delta_k^x\|^2 + \|\Delta_k^y\|^2$, as well as for the constraint criticality measure, $\|\nabla h(x_k, y_k)\|^2$. This serves as a preliminary result for the first part of our analysis. In Corollary 4.3, we further show that, under a regularity assumption, these bounds yield a convergence guarantee for finding an $\epsilon$-KKT point of problem 3 as stated in Definition 2.4.

**Theorem 4.2.** *Suppose that Assumptions 2.1 and 2.2 hold and $\rho(x, y) = \|\nabla h(x, y)\|^2$. Let $\{(x_k, y_k, \lambda_k)\}_{k=0}^{K-1}$ be the sequence generated by Algorithm 1 with $C_0 > 0$ and step size $\gamma > 0$ such that $\gamma \le \min\{\alpha, \frac{1}{L_f + \alpha L_h}\}$. Then for all $K \ge 1$,*

$$
\frac{1}{K} \sum_{k=0}^{K-1} \big(\|\Delta_k^x\|^2 + \|\Delta_k^y\|^2\big) \le \frac{4(f_0 + \alpha^3 C_0 - \bar{f})}{\gamma K} + \frac{2\alpha C_0}{\gamma L_h K} + 2\alpha^2 L_h C_f^2,
$$

$$
\frac{1}{K} \sum_{k=0}^{K-1} \big(\|\nabla_x h(x_k, y_k)\|^2 + \|\nabla_y h(x_k, y_k)\|^2\big) \le \frac{2\alpha C_0}{\gamma K} + \frac{2L_h(f_0 + \alpha^3 C_0 - \bar{f})}{\gamma K} + \alpha^2 L_h^2 C_f^2
$$

*Proof.* For the sake of simplicity throughout the remainder of the proofs, we define the vectors $z_k = (x_k, y_k)$ and $\Delta_k = (\Delta_k^x, \Delta_k^y)$, both belonging to $\mathbb{R}^{n+m}$.

Function $f$ is continuously differentiable with a Lipschitz continuous gradient, characterized by $L_f = \max\{L_x^f, L_y^f\}$. Using the smoothness of the function $f$, we have

$$
\begin{aligned}
f(z_{k+1}) &\le f(z_k) + \nabla f(z_k)^\top (z_{k+1} - z_k) + \frac{L_f}{2}\|z_{k+1} - z_k\|^2 = f(z_k) + \gamma \nabla f(z_k)^\top \Delta_k + \frac{\gamma^2 L_f}{2}\|\Delta_k\|^2 \\
&= f(z_k) + \gamma(\nabla f(z_k) + \Delta_k)^\top \Delta_k + (\frac{\gamma^2 L_f}{2} - \gamma)\|\Delta_k\|^2 = f(z_k) - \gamma\lambda(z_k)\nabla h(z_k)^\top \Delta_k + (\frac{\gamma^2 L_f}{2} - \gamma)\|\Delta_k\|^2. \quad (9)
\end{aligned}
$$

Since $(\Delta_k, \lambda(z_k))$ is the optimal primal-dual pair for the subproblem in equation 7 at iteration $k$, the complementarity slackness condition implies that $\lambda(z_k)(\nabla h(z_k)^\top \Delta_k + \alpha\|\nabla h(z_k)\|^2) = 0$. Using this result within the above inequality, we obtain

$$
f(z_{k+1}) - f(z_k) \le (\frac{\gamma^2 L_f}{2} - \gamma)\|\Delta_k\|^2 + \gamma\alpha\lambda(z_k)\|\nabla h(z_k)\|^2. \qquad (10)
$$

Similarly, using the smoothness of function $h$ and the update of $\Delta_k$, we have

$$
\begin{aligned}
h(z_{k+1}) - h(z_k) &\le \gamma \nabla h(z_k)^\top \Delta_k + \frac{\gamma^2 L_h}{2}\|\Delta_k\|^2 = -\gamma \nabla h(z_k)^\top \nabla f(z_k) - \gamma\lambda(z_k)\|\nabla h(z_k)\|^2 + \frac{\gamma^2 L_h}{2}\|\Delta_k\|^2 \\
&\le \frac{\gamma}{2\alpha L_h}\|\nabla h(z_k)\|^2 + \frac{\alpha\gamma L_h}{2}C_f^2 - \gamma\lambda(z_k)\|\nabla h(z_k)\|^2 + \frac{\gamma^2 L_h}{2}\|\Delta_k\|^2, \qquad (11)
\end{aligned}
$$

where in the last inequality we used Young's inequality $-\nabla f(z_k)^T \nabla h(z_k) \le \frac{\alpha L_h}{2}\|\nabla f(z_k)\|_2^2 + \frac{1}{2\alpha L_h}\|\nabla h(z_k)\|_2^2$ for $\alpha > 0$ as well as the boundedness of $\nabla f$. Let us define $v(z) \triangleq f(z) + \alpha h(z)$. Combining the above

inequalities by multiplying equation 11 with $\alpha$ and adding to equation 10 we obtain

$$v(z_{k+1}) - v(z_k) \leq \frac{\gamma}{2L_h}\|\nabla h(z_k)\|^2 + \frac{\alpha^2\gamma L_h}{2}C_f^2 + \gamma(\frac{L_f + \alpha L_h}{2}\gamma - 1)\|\Delta_k\|^2.$$

Next, assuming that $\gamma \leq \frac{1}{L_f + \alpha L_h}$ and rearranging the terms we conclude that

$$\frac{1}{2}\|\Delta_k\|^2 \leq \frac{v(z_k) - v(z_{k+1})}{\gamma} + \frac{1}{2L_h}\|\nabla h(z_k)\|^2 + \frac{\alpha^2 L_h}{2}C_f^2. \tag{12}$$

On the other hand, using the smoothness of $h$ once again along with the fact that $\Delta_k$ is a feasible point of equation 7, we have

$$h(z_{k+1}) - h(z_k) \leq \gamma\nabla h(z_k)^\top\Delta_k + \frac{\gamma^2 L_h}{2}\|\Delta_k\|^2 \leq -\gamma\alpha(\|\nabla h(z_k)\|^2) + \frac{\gamma^2 L_h}{2}\|\Delta_k\|^2.$$

Rearranging the terms leads to

$$\frac{1}{2L_h}\|\nabla h(z_k)\|^2 \leq \frac{h(z_k) - h(z_{k+1})}{2\alpha\gamma L_h} + \frac{\gamma}{4\alpha}\|\Delta_k\|^2. \tag{13}$$

Now adding up equation 12 and equation 13 and using $\gamma \leq \alpha$, we obtain

$$\frac{1}{4}\|\Delta_k\|^2 \leq \frac{v(z_k) - v(z_{k+1})}{\gamma} + \frac{h(z_k) - h(z_{k+1})}{2\alpha\gamma L_h} + \frac{\alpha^2 L_h}{2}C_f^2.$$

Summing the above inequality over $k = 0$ to $K - 1$ and divide both sides $K/4$ leads to

$$\frac{1}{K}\sum_{k=0}^{K-1}\|\Delta_k\|^2 \leq \frac{4(v(z_0) - v(z_K))}{\gamma K} + \frac{2(h(z_0) - h(z_K))}{\alpha\gamma L_h K} + 2\alpha^2 L_h C_f^2$$

$$\leq \frac{4(f(z_0) + \alpha^3 C_0 - \bar{f})}{\gamma K} + \frac{2\alpha C_0}{\gamma L_h K} + 2\alpha^2 L_h C_f^2, \tag{14}$$

where in the last inequality we used nonegativity of function $h$, the lower-bound on function $f$, and the initialization condition $h(z_0) \leq \alpha^2 C_0$. Furthermore, summing the result in equation 13 over $k = 0$ to $K - 1$ and dividing both sides by $K/(2L_h)$ and using equation 14 implies the desired result as follows

$$\frac{1}{K}\sum_{k=0}^{K-1}\|\nabla h(z_k)\|^2 \leq \frac{h(z_0) - h(z_K)}{\alpha\gamma K} + \frac{L_h\gamma}{2\alpha K}\sum_{k=0}^{K-1}\|\Delta_k\|^2 \leq \frac{2\alpha C_0}{\gamma K} + \frac{2L_h(f(z_0) + \alpha^3 C_0 - \bar{f})}{\gamma K} + \alpha^2 L_h^2 C_f^2. \tag{15}$$

$\square$

**Corollary 4.3.** *Suppose the following regularity assumption hold: there exists $c > 0$ such that $\|\nabla_y g(x,y)\| \leq c\|\nabla h(x,y)\|$ for any $(x,y)$. Let $\nu_k = \lambda_k\nabla_y g(x_k, y_k)$ for any $k \geq 0$, then there exists $B > 0$ such that $\|\nu_k\| \leq B$ for any $k \geq 0$. Moreover, under the premises of Theorem 4.2 there exists, $t \in \{0, \ldots, K - 1\}$ such that $(x_t, y_t, \nu_t) \in \mathbb{R}^n \times \mathbb{R}^m \times \mathbb{R}^m$ is an $\epsilon$-KKT point of problem equation 3, i.e,*

$$\max\left\{\|\nabla_y g(x_t, y_t)\|^2, \|\nabla f(x_t, y_t) + [\nabla_{yx}g(x_t, y_t) \ \nabla_{yy}g(x_t, y_t)]^\top\nu_t\|^2\right\} \leq \epsilon,$$

*within $K = \mathcal{O}(1/\epsilon^{1.5})$ iterations.*

*Proof.* The proof is provided in the Appendix section A. $\square$

**Remark 4.4.** The regularity condition used in Corollary 4.3 is equivalent to the function $h(x,y) = \|\nabla_y g(x,y)\|^2$ satisfying the PL condition, which is weaker than $g(x, \cdot)$ being strongly convex yet more restrictive than merely assuming that $g$ satisfies PL condition. However, this is a commonly used regularity condition in the optimization literature and has been employed in the convergence analysis of iterative

algorithms for nonconvex optimization problems with functional constraints (Bolte et al., 2018; Sahin et al., 2019; Li et al., 2021; Lin et al., 2022; Lu, 2022; Li et al., 2024) corresponding to equation 3. The properties and practical relevance of this assumption have been extensively examined in (Bolte et al., 2018; Sahin et al., 2019), where it is shown to be a weaker condition than the Mangasarian-Fromovitz Constraint Qualification (MFCQ) when $h(x, y)$ has a minimizer. In the context of bilevel optimization, a notable example of a lower-level function $g$ that satisfies this regularity assumption is $g(x, y) = \frac{1}{2}\|Ay - Bx\|^2$, where $A \in \mathbb{R}^{p \times m}$ and $B \in \mathbb{R}^{p \times n}$ which arises as a loss function in various applications, including robust and adversarial learning.

While the considered regularity assumption in Corollary 4.3 leads to a favorable convergence rate for finding an $\epsilon$-KKT point, it imposes limitations on the class of optimization problems to which our proposed algorithm can effectively be applied. In response to this limitation, we next show that by selecting a different function $\rho$, we can eliminate the need for the stringent regularity assumption. To this end, once again, we first demonstrate some preliminary convergence bounds which help us to obtain a convergence guarantee for finding an $\epsilon$-KKT point of problem equation 3 in Corollary 4.6.

**Theorem 4.5.** *Suppose that Assumptions 2.1 and 2.2 hold and consider $\rho(x, y) = \|\nabla h(x, y)\|\sqrt{h(x_0, y_0)}$. Let $\{(x_k, y_k, \lambda_k)\}_{k=0}^{K-1}$ be the sequence generated by Algorithm 1 with $C_0 > 0$ and stepsize $\gamma > 0$ such that $\gamma = \min\{\frac{1}{K^{2/3}}, \frac{1}{L_f}\}$. Define $B_\Delta \triangleq 2C_f + \alpha^2\sqrt{C_0}$, then for all $K \geq 1$,*

$$\frac{1}{K}\sum_{k=0}^{K-1} h(x_k, y_k) \leq \alpha^2 C_0 + \frac{\gamma^2 L_h B_\Delta^2}{2}\frac{(K-1)}{2}, \tag{16a}$$

$$\frac{1}{K}\sum_{k=0}^{K-1}\left(\|\Delta_k^x\|^2 + \|\Delta_k^y\|^2\right) \leq \frac{2(f_0 - \bar{f})}{\gamma K} + \alpha^2(B_\Delta^2 + C_0). \tag{16b}$$

*Proof.* Using the smoothness of $h$ again along with the fact that $\Delta_k$ is a feasible point of equation 7 we have

$$h(z_{k+1}) - h(z_k) \leq \gamma \nabla h(z_k)^\top \Delta_k + \frac{\gamma^2 L_h}{2}\|\Delta_k\|^2 \leq -\gamma\alpha(\|\nabla h(z_k)\|(h(z_0))^{1/2}) + \frac{\gamma^2 L_h}{2}\|\Delta_k\|^2. \tag{17}$$

Note that $\Delta_k$ can be bounded as follows:

$$\|\Delta_k\| \leq \|\nabla f(z_k)\| + \lambda(z_k)\|\nabla h(z_k)\| \leq B_\Delta \triangleq 2C_f + \alpha^2\sqrt{C_0}. \tag{18}$$

Then, from equation 17 and using the above bound followed by a telescopic summation, we have that for any $k \geq 0$ $h(z_k) \leq h(z_0) + \frac{\gamma^2 L_h}{2}B_\Delta^2 k$. Taking the average of the above inequality from $k = 0$ to $K - 1$ and using the initialization condition, we obtain $\frac{1}{K}\sum_{k=0}^{K-1} h(z_k) \leq \alpha^2 C_0 + \frac{\gamma^2 L_h B_\Delta^2}{2}\frac{(K-1)}{2}$ which proves equation 16a. Moreover, similar to the proof of the previous result, we can show that

$$f(z_{k+1}) - f(z_k) \leq (\frac{\gamma^2 L_f}{2} - \gamma)\|\Delta_k\|^2 + \gamma\alpha\lambda(z_k)\|\nabla h(z_k)\|(h(z_0))^{1/2}$$

$$\leq (\frac{\gamma^2 L_f}{2} - \gamma)\|\Delta_k\|^2 + \gamma\alpha B_\Delta(h(z_0))^{1/2} \leq (\frac{\gamma^2 L_f}{2} - \gamma)\|\Delta_k\|^2 + \frac{\gamma\alpha^2}{2}B_\Delta^2 + \frac{\gamma}{2}h(z_0). \tag{19}$$

Now, selecting $\gamma \leq 1/L_f$, rearranging the terms and taking average over $k = 0$ to $K - 1$ we obtain

$$\frac{1}{K}\sum_{k=0}^{K-1}\|\Delta_k\|^2 \leq \frac{2(f_0 - f_K)}{\gamma K} + \alpha^2 B_\Delta^2 + \frac{1}{K}\sum_{k=0}^{K-1} h(z_0) \leq \frac{2(f_0 - f_K)}{\gamma K} + \alpha^2 B_\Delta^2 + \alpha^2 C_0, \tag{20}$$

which, together with the assumption of the lower bound on the upper-level objective, yields equation 16b. $\square$

**Corollary 4.6.** *Let $\{(x_k, y_k, \lambda_k)\}_{k=0}^{K-1}$ be the sequence generated by Algorithm 1 with $\alpha = K^{-1/6}$ and $\gamma = \min\{\frac{1}{K^{2/3}}, \frac{1}{L_f}\}$. Under the premises of Theorem 4.5 we have that $\frac{1}{K}\sum_{k=0}^{K-1}\|\Delta_k^x\|^2 + \|\Delta_k^y\|^2 \leq \mathcal{O}(1/K^{1/3})$ and $\frac{1}{K}\sum_{k=0}^{K-1} h_k \leq \mathcal{O}(1/K^{1/3})$. Let $\nu_k = \lambda_k\nabla_y g(x_k, y_k)$ for any $k \geq 0$, then there exists $t \in \{0, \ldots, K-1\}$ such that $(x_t, y_t, \nu_t)$ is an $\epsilon$-KKT point of problem 3, i.e.,*

$$\max\left\{\|\nabla_y g(x_t, y_t)\|^2, \|\nabla f(x_t, y_t) + [\nabla_{yx} g(x_t, y_t) \ \nabla_{yy} g(x_t, y_t)]^\top \nu_t\|^2\right\} \leq \epsilon,$$

*within $K = \mathcal{O}(1/\epsilon^3)$ iterations.*

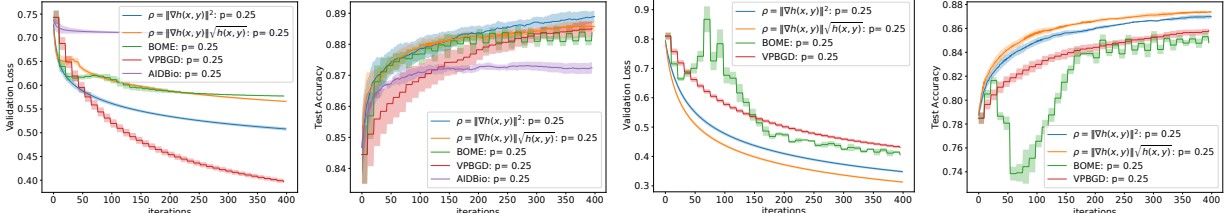

Figure 2: Comparison between our method and the state of the art on the DHC benchmark with corruption rate $p = 25\%$. The first two plots from left show the validation loss and the accuracy of the test set on the DHC benchmark with PCA. The last two plots show the validation loss and the accuracy of the test set on the large-scale DHC problem.

*Proof.* The proof is provided in the Appendix section B. $\qquad\square$

## 5 Numerical Experiments

In this section, we numerically evaluate the performance of our method compared to other methods to solve two instances of the Data Hyper-cleaning (DHC) problem on the MNIST (LeCun et al., 2010) dataset. We compare our methods with Bome (Liu et al., 2022) and VPBGD (Shen & Chen, 2023) across all experimental settings, and with AIDBio (Ji et al., 2021) only in a small-scale, strongly convex setup due to the high computational cost of calculating Hessians and the fact that AIDBio was designed specifically for strongly convex problems. We set the parameters according to the values suggested in their theoretical analyses, and compare performance in terms of validation loss and test accuracy as functions of the number of iterations. The results in this section are averaged over 5 runs with different seeds. Additional numerical experiments on synthetic problems are provided in Appendix section C. The code implementations can be found here.

**Data Hyper-Cleaning**

Consider a DHC problem, where some of the labels in the training data have been corrupted, and the goal is to train a classifier utilizing the clean validation data. In data hyper-cleaning, the objective is to automatically assign weights to the training samples such that mislabeled or unreliable data points receive lower importance during training. These weights are optimized through the following bilevel formulation, where the upper level minimizes the validation loss, and the lower level corresponds to training on the weighted training data.

$$\min_x \frac{1}{N_{\text{val}}} \sum_{(a_i, b_i) \in \mathcal{D}_{\text{val}}} \mathcal{L}(a_i^\top y^\star(x), b_i) \quad \text{s.t.} \quad y^\star(x) = \arg\min_y \frac{1}{N_{\text{tr}}} \sum_{(a_i, b_i) \in \mathcal{D}_{\text{tr}}} \sigma(x_i)\mathcal{L}(a_i^\top y, b_i) + \lambda \|y\|^2,$$

where $\lambda = 0.001$ is the regularizer and $\sigma(\cdot)$ and $\mathcal{L}(\cdot)$ represent the sigmoid function and cross-entropy loss, respectively. The upper-level variable $x \in \mathbb{R}^{5000}$ represents the sample weights, and the lower-level variable $y$ is the weight of the classifier. We select 10000 samples and split them into 5000 for training ($\mathcal{D}_{\text{tr}}$), 2500 for validation ($\mathcal{D}_{\text{val}}$), and 2500 for testing ($\mathcal{D}_{\text{test}}$). We then run the experiment under two setups, one that reduces the dimensionality of the problem using Principal Component Analysis (PCA), and the other one that tests our method in a high-dimensional setting.

**Low-dimension DHC:** We first use PCA to reduce the dimensions of the problem to $y \in \mathbb{R}^{82 \times 10}$ and run the experiment with corruption rate $p = 25\%$. The first two plots in Fig. 2 compare our methods with VPBGD, BOME, and AIDBiO in terms of validation loss and test accuracy, where we observe faster convergence of our method in terms of test accuracy.

**High-dimension DHC:** In this experiment, we aim to study the performance of our method in high-dimensional benchmarks. Therefore, we do not use PCA, which translates to $y$ being a $784 \times 10$ matrix. The final two plots in Fig. 2 compare our methods with VPBGD and BOME in terms of validation loss and test accuracy, where we observe a significantly faster convergence of our method, showcasing the effectiveness of our approach in large-scale settings.

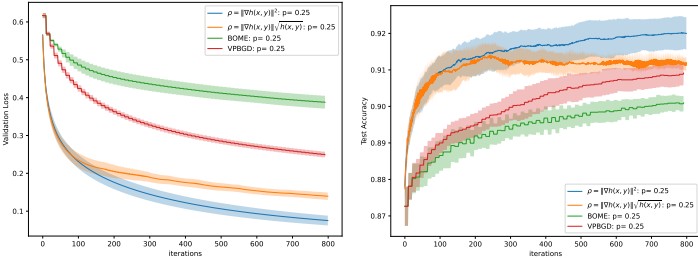

Figure 3: Comparisons of the validation loss and test accuracy between our method from Theorem 4.2 and BOME on the DHC problem with the neural network classifier.

**Neural Network (NN) Classifier:** To evaluate our method on yet another large-scale nonconvex method, we use a fully connected NN with one hidden layer and ReLU activations to solve the DHC problem.

$$\min_x \frac{1}{N_{\text{val}}} \sum_{(a_i,b_i)\in\mathcal{D}_{\text{val}}} \mathcal{L}(f_{y^\star(x)}(a_i), b_i) \quad \text{s.t.} \quad y^\star(x) = \arg\min_y \frac{1}{N_{\text{tr}}} \sum_{(a_i,b_i)\in\mathcal{D}_{\text{tr}}} \sigma(x_i)\mathcal{L}(f_y(a_i), b_i) + \lambda\|y\|^2,$$

where $f_\theta(.)$ denotes the NN parameterized by $\theta$. Fig. 3 shows that our method outperformed BOME and VPBGD in both validation loss and test accuracy. VPBGD parameters were chosen from their GitHub[1].

# 6 Limitations & Future Works

Despite strong theoretical and empirical results, the proposed method has several limitations. A primary issue is the computational cost of Hessian-related operations required to enforce the lower-level optimality condition. In particular, our method relies on Hessian-vector products (see Remark 3.1), which, although efficiently implemented via automatic differentiation, can still be expensive in high-dimensional settings, thereby limiting scalability compared to fully first-order methods. Developing more efficient approximations of these operations remains an important direction for future work. Another important future direction is to extend the proposed framework to constrained bilevel optimization. In general, the problem of constrained bilevel optimization has been studied extensively Dempe & Zemkoho (2012); Khanduri et al. (2023); Dempe & Mehlitz (2025) and more. The presence of lower-level constraints introduces additional challenges, making the problem significantly harder to solve. For instance, when considering lower-level constraints, the lower-level optimality conditions are no longer given simply by $\nabla_y g(x,y) = 0$; instead, they become a set of KKT conditions. This leads to the so-called KKT reformulation of bilevel optimization, which transforms the problem into a mathematical program with complementarity constraints (MPCC). It has been shown that this reformulation is generally not equivalent to the original problem Dempe & Dutta (2012), even if the lower-level problem (both objective and constraints) is convex. The reason is that the complementary slackness condition violates standard constraint qualifications (such as LICQ or MFCQ) at every feasible point. Consequently, the first-order necessary condition, namely that any solution of the bilevel problem satisfies the KKT conditions, may no longer hold. Due to these challenges, we defer analyses of bilevel optimization problems with lower-level constraints to future work.

# 7 Conclusion

In this paper, we proposed an inversion-free single-time scale method to solve the bilevel optimization problem of the form BLO. Our idea hinged on the use of gradient descent to decrease the upper-level objective, coupled with a convex QP that minimally perturbed the gradient descent directions to reduce the sub-optimality of the condition imposed by the lower-level problem. We proposed two methods, one assumed a certain regularity condition and guaranteed to find a stationary point with an iteration complexity of $\mathcal{O}(1/\epsilon^{1.5})$, and the other one relaxed the assumption and, in turn, proved a complexity of $\mathcal{O}(1/\epsilon^3)$. Furthermore, we ran extensive numerical analysis, showcasing the performance of our methods against state-of-the-art and under convex and nonconvex settings.

---

[1]https://github.com/hanshen95/penalized-bilevel-gradient-descent/

## Acknowledgment

This work was supported in part by the National Science Foundation under Grants 2515978 and 2515979. Any opinions, findings, and conclusions or recommendations expressed in this material are those of the authors and do not necessarily reflect the views of the National Science Foundation.

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

## A Proof of Corollary 4.3

*Proof.* First note that based on the definition of $\lambda_k$ in equation 8b and the regularity assumption we have that $\|\nu_k\| = |\lambda_k| \|\nabla_y g(x_k, y_k)\| = \frac{\|\nabla f(x_k, y_k)\|}{\|\nabla h(x_k, y_k)\|} \|\nabla_y g(x_k, y_k)\| + \alpha \|\nabla_y g(x_k, y_k)\| \leq c \|\nabla f(x_k, y_k)\| + \alpha \|\nabla_y g(x_k, y_k)\| \leq c\, C_f + \alpha\, C_g$ where the last inequality follows from the Assumptions 2.1 and 2.2. This implies that $\|\nu_k\|$ is bounded.

Suppose $\rho(x, y) = \|\nabla h(x, y)\|^2$, using the result of Theorem 4.2 combined with the regularity condition, we have the following convergence bounds

$$\frac{1}{K} \sum_{k=0}^{K-1} \left( \|\Delta_k^x\|^2 + \|\Delta_k^y\|^2 \right) \leq \frac{4(f_0 + C_0 - \bar{f})}{\gamma K} + \frac{2C_0}{\gamma L_h K} + 2\alpha^2 L_h C_f^2,$$

and,

$$\frac{1}{cK} \sum_{k=0}^{K-1} \|\nabla_y g(x_k, y_k)\|^2 \overset{\text{regularity}}{\leq} \frac{1}{K} \sum_{k=0}^{K-1} \|\nabla h(x_k, y_k)\|^2 \leq \frac{2C_0}{\gamma K} + \alpha^2 L_h^2 C_f^2 + \frac{2L_h(f_0 + C_0 - \bar{f})}{\gamma K}.$$

Setting $\alpha = K^{-1/3}$ and $\gamma = \min\{\alpha, \frac{1}{L_f + \alpha L_h}\}$, implies that $\gamma = \Omega(1/K^{1/3})$. Substituting $\alpha$ and $\gamma$ we obtain $\frac{1}{K} \sum_{k=0}^{K-1}(\|\Delta_k^x\|^2 + \|\Delta_k^y\|^2) = \mathcal{O}(K^{-2/3})$, and $\frac{1}{K} \sum_{k=0}^{K-1} \|\nabla_y g(x_k, y_k)\|^2 = \mathcal{O}(K^{-2/3})$.

Now, recall that $\nu_k = \lambda_k \nabla_y g(x_k, y_k)$ for any $k \geq 0$ and let $t \triangleq \arg\min_{0 \leq k \leq K-1}\{\max\{\|\Delta_k\|^2, \|\nabla_y g(x_k, y_k)\|^2\}\}$, then we conclude that $(\|\Delta_t^x\|^2 + \|\Delta_t^y\|^2) = \|\nabla f(x_t, y_t) + [\nabla_{yx} g(x_t, y_t) \ \nabla_{yy} g(x_t, y_t)]^\top \nu_t\|^2 \leq \epsilon$ and $\|\nabla_y g(x_t, y_t)\|^2 \leq \epsilon$ after $K = \mathcal{O}(1/\epsilon^{1.5})$ iterations. $\square$

## B Proof of Corollary 4.6

*Proof.* Suppose $\rho(x, y) = \|\nabla h(x, y)\| \sqrt{h(x_0, y_0)}$, using the result of Theorem 4.5, we have the following convergence bounds

$$\frac{1}{K} \sum_{k=0}^{K-1} \|\Delta_k\|^2 \leq \frac{2(f_0 - f_K)}{\gamma K} + \alpha^2 B_\Delta^2 + \alpha^2 C_0,$$

$$\frac{1}{K} \sum_{k=0}^{K-1} h(x_k, y_k) \leq \alpha^2 C_0 + \frac{\gamma^2 L_h B_\Delta^2 (K-1)}{4}.$$

Setting $\alpha = K^{-1/6}$ and $\gamma = \min\{\frac{1}{K^{2/3}}, \frac{1}{L_f}\}$ implies that $\gamma = \Omega(1/K^{2/3})$. Substituting $\alpha$ and $\gamma$ into these inequalities simplifies them to $\frac{1}{K} \sum_{k=0}^{K-1} \|\Delta_k\|^2 = \mathcal{O}(K^{-1/3})$ and $\frac{1}{K} \sum_{k=0}^{K-1} h(x_k, y_k) = \mathcal{O}(K^{-1/3})$.

Now, let $t \triangleq \arg\min_{0 \leq k \leq K-1}\{\max\{\|\Delta_k\|^2, h(x_k, y_k)\}\}$, then we conclude that $\|\Delta_t\|^2 \leq \mathcal{O}(K^{-1/3})$ and $h(x_t, y_t) \leq \mathcal{O}(K^{-1/3})$. Utilizing the definition of $h(x, y) = \|\nabla_y g(x, y)\|^2$, we directly obtain $\|\nabla_y g(x_t, y_t)\|^2 \leq \mathcal{O}(1/K^{1/3})$. Hence, letting $\nu_k = \lambda_k \nabla_y g(x_k, y_k)$ we conclude that achieving $\|\nabla_y g(x_t, y_t)\|^2 \leq \epsilon$ and $\|[\nabla_{yx} g(x_t, y_t) \ \nabla_{yy} g(x_t, y_t)]^\top \nu_t\|^2 \leq \epsilon$, requires $1/K^{1/3} \leq \epsilon$, which leads to $K = \mathcal{O}(1/\epsilon^3)$. Consequently, our proposed algorithm can achieve an $\epsilon$-KKT point $(x_t, y_t, \nu_t)$ within $\mathcal{O}(1/\epsilon^3)$ iterations. $\square$

## C Additional Numerical Experiments

In this section, we numerically evaluate the performance of our method compared with other methods, including BOME (Liu et al., 2022), VPBGD (Shen & Chen, 2023), and AIDBiO (Ji et al., 2021), on some synthetic problems with strongly convex and nonconvex lower-level problems. For our methods, we include the result for different choices for $\alpha$ and $\gamma$ from Theorem 4.2 and Theorem 4.5. We compare the norm of the hyper-gradient ($\|F(x_k, y_k)\|$) as the metric for convergence. However, for the experiments where the lower-level function is not strongly convex, we compare $\|\Delta_k\|$ since the hyper-gradient might not be well-defined.

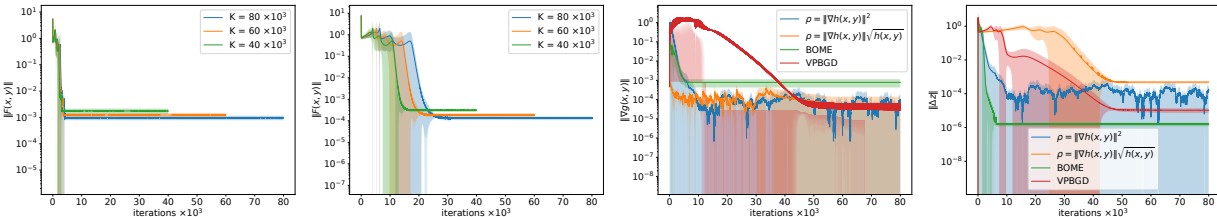

Figure 4: Effect of the number of iterations on the convergence of the strongly convex and nonconvex synthetic examples. Parameter choices are based on *(leftmost)* Theorem 4.2 and *(mid-left)* Theorem 4.5 for the strongly convex synthetic example. The *(mid-right)* and *(rightmost)* panels present a comparison with BOME (Liu et al., 2022) on the synthetic example with a nonconvex lower-level function.

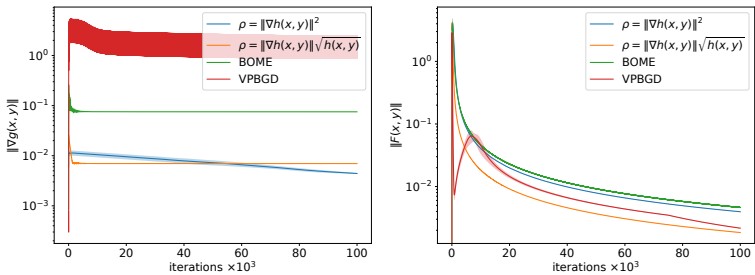

Figure 5: Comparison of our method, AIDBiO, and BOME on the coreset selection problem.

## C.1 Synthetic Example

To showcase the performance of our method, we start with two simple numerical examples.

### C.1.1 Strongly Convex Lower-level

Consider the following basic bilevel optimization problem

$$\min_x \quad \sin(c^\top x + d^\top y^\star(x)) + \log(\|x + y^\star(x)\|^2 + 1)$$
$$\text{s.t.} \quad y^\star(x) \in \arg\min_y \tfrac{1}{2}\|Hy - x\|^2,$$

where $x, y, c, d \in \mathbb{R}^{20}$ and $H \in \mathbb{R}^{20 \times 20}$ is randomly generated in a way such that its condition number is no larger than 10. The setup is similar to the setup of Sharifi et al. (2025).

The first two plots in Fig. 4 show the reduction in the norm of the hyper-gradient with respect to the number of iterations using parameter choices from both Theorem 4.2 and Theorem 4.5. In both cases, as we expect, our proposed algorithm converges to a more accurate solution as we increase the number of iterations, even though it takes longer to converge.

### C.1.2 Coreset Selection

Following (Liu et al., 2022), we consider the following coreset selection problem, which is a bilevel optimization problem with a strongly convex lower-level function,

$$\min_x \quad \|y^\star(x) - y_0\|_2^2 \quad \text{s.t.} \quad y^\star(x) \in \arg\min_y \|y - A\sigma(x)\|_2^2$$

where $\sigma(x) = \exp(x)/\sum_{i=1}^4 \exp(x_i)$ is the softmax function, $x \in \mathbb{R}^4, y \in \mathbb{R}^2$ and $A \in \mathbb{R}^{2 \times 4}$. We compare our method against VPBGD, BOME, and AIDBiO. The results are depicted in Fig. 5, which shows that our method from Theorem 4.5 converges faster than the rest of the methods.

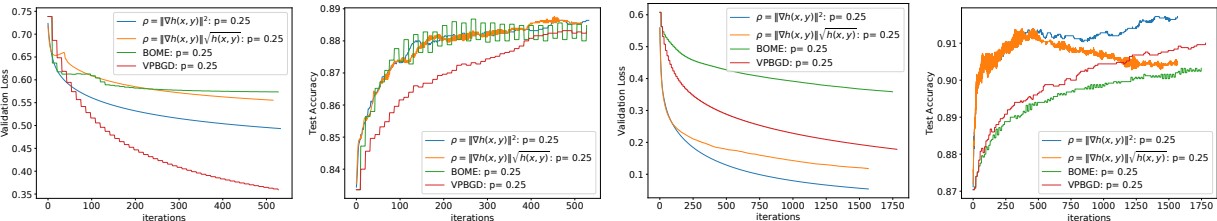

Figure 6: Time comparison of our method, VPBGD, and BOME on the neural network DHC problem.

### C.1.3   Nonconvex Lower-level

To also test our method on a benchmark with a nonconvex lower-level problem, we slightly change the previous setup and design a lower-level problem such that it is nonconvex. In particular, consider the following bilevel optimization problem

$$\min_x \quad \sin(c^\top x + d^\top y^\star(x)) + \log(\|x + y^\star(x)\|^2 + 1)$$
$$\text{s.t.} \quad y^\star(x) \in \arg\min_y \cos(\tfrac{1}{2}\|Hy - x\|^2),$$

where the parameters are generated in the same manner as in the previous subsection.

The last two plots in Fig. 4 show the comparison between our method and BOME (Liu et al., 2022), showcasing that our method remains close to the lower-level optimal point $y^\star(x)$, while reducing the KKT stationary condition.

### C.2   Time Comparison

In the experimental section, we used a fixed number of iterations to compare the different algorithms. This raises a natural question about fairness, since the compared methods range from single-loop to double-loop schemes and from first-order to second-order methods.

To address this concern, we provide all methods with a fixed time budget and report the results in Figure 6. Specifically, we evaluate the methods on the DHC problem with PCA (to illustrate performance in a small-scale setting) using a time budget of 250 seconds, and on the same problem with a neural network classifier (to demonstrate behavior in a large-scale setting) using a time budget of 1000 seconds. We observe that all methods run within the same order of computational time, completing nearly the same number of iterations in the DHC-with-PCA setup. In the neural network setting, our method is only slightly slower than the other two methods, owing to the PyTorch computation trick described in Remark 3.1.

