# OpenReview forum: "Perturbed Gradient Descent via Convex Quadratic Approximation for Nonconvex Bilevel Optimization"
_TMLR — Accepted by TMLR_

### Review · Reviewer_2ChR · 2025-11-11

**Summary Of Contributions:**

This paper proposes a new gradient-based method for solving nonconvex bilevel optimization problems. The method introduces a convex Quadratic Program (QP) at each iteration that perturbs the gradient descent directions to mitigate suboptimality arising from the lower-level problem. The authors provide convergence guarantees under two settings: with and without a regularity assumption (PL condition), achieving iteration complexities of $\mathcal{O}(\epsilon^{-1.5})$ and $\mathcal{O}(\epsilon^{-3})$, respectively. The proposed algorithm is inversion-free and evaluated on data hyper-cleaning tasks, demonstrating competitive performance.

**Additional Comments:**

Please see above.

**Audience:**

Yes

**Audience Explanation:**

The bilevel optimization problem has wide application in engineering, economics, and machine learning. They propose a new gradient-based method for solving nonconvex bilevel optimization problems.

**Claims And Evidence:**

Yes

**Claims Explanation:**

Yes. I think the statements and proof are right.

**Requested Changes:**

* Clarification of the QP formulation (Eq. 5):
More explanation would be helpful when introducing the convex QP (5). As it is central to the algorithm, elaborating on its intuition and how the constraint enforces progress on the lower-level condition would improve clarity.

* Notation consistency for $h$:
The definition of $h$ is confusing. In Eq. (4),  $h(x,y)=||\nabla_y g(x,y)||^2$, whereas in Remark 3.1, $h(x,y)=\nabla_y g(x,y)$ seems to be used. Please ensure consistent notation throughout the paper.

* Minor typo and consistency:
In Assumption 2.2, the upper bound for $||\nabla_y g(x,y)||$ is denoted by $C_g$, but in Lemma 4.1, it appears as $C_y^g$. It would be clearer to unify these symbols.

* Dependence on the unconstrained lower-level problem:
The convergence analysis appears to rely on the unconstrained nature of the lower-level problem. Would it be possible to extend the results to constrained bilevel settings for both upper and lower levels? With compact constraint sets, the boundedness assumptions (Assumptions 2.1-2.2) would become more natural and easier to justify.

* Improvement under convexity:
Can the convergence results be improved when the lower-level problem is (purely) convex? A discussion or brief remark on this special case would be informative.

* Choice of $\rho_k$:
The rationale behind the choice of $\rho_k$ could be better motivated. Please provide an intuitive explanation of how these specific choices help relax the regularity condition and influence the convergence dynamics.

* Stationary measure and comparison to prior work:
It would be helpful to justify the definition of the stationary measure used in your analysis. When the lower-level function satisfies the PL condition, how does your notion of an $\epsilon$-KKT point relate to those used in prior bilevel optimization literature?

---

> ### Author Response · Authors · 2026-02-17
> **Rebuttal of Review 2ChR**
>
> **R1.**
> In the revised manuscript, we have expanded the explanation of the convex QP in Eq. (5) to clarify its role and intuition. Specifically, we now describe how the QP selects velocity vectors $ (\dot{x}, \dot{y}) $ that remain as close as possible to the nominal gradient-descent direction on the upper-level objective while satisfying the linear constraint $\dot{h} + \alpha h = 0 .$
> We further explain that this constraint admits the solution $ h(t) = h(0)e^{-\alpha t} $, thereby ensuring exponential decay of $ h $ toward satisfying the lower-level condition. This additional explanation appears right after Eq. (5).
>
> **R2.**
> Thank you for bringing this issue to our attention. We would like to clarify that Remark 3.1 uses the derivatives of the function $h$, which, based on our definition in equation (4), should be modified to include the constant ``2'' as follows:
> $\nabla_x h = 2\nabla_{yx}^2 g^\top  \nabla_y g,  \quad \nabla_y h = 2\nabla_{yy}^2 g^\top  \nabla_y g.$
>
> **R3.**
>  We appreciate your attention to the details. The typos have been fixed in the revised manuscript.
>
> **R4.**
> Thanks for your great questions. Introducing upper- and lower-level constraint sets leads to two distinct difficulties that deserve future work. First, if we consider a convex constraint set $X$ for the upper-level problem, i.e., $\min_{x\in X,y} f(x,y) \text{s.t.} h(x,y)=0$, then the existence of a KKT solution requires imposing a suitable constraint qualification that accounts for all constraints, both those defining the set $X$ and the nonlinear constraint $h$. Furthermore, the QP subproblem must be modified, potentially to a convex program:
>
> $$
> (\Delta_k^x,\Delta_k^y)=\arg\min_{\Delta^x,\Delta^y}\frac{1}{2}||\Delta^x+\nabla_x f_k||^2_2+\frac{1}{2}||\Delta^y+\nabla_y f_k||^2_2
> $$
> $$
> \hbox{s.t.}\quad \nabla_x h_k^\top \Delta^x + \nabla_y h_k^\top \Delta^y + \alpha \rho_k \leq 0,\quad x_k+\gamma \Delta^x\in X. \notag
> $$
>
> This subproblem can be equivalently rewritten in terms of the next iterates:
> $$
> (x_{k+1},y_{k+1})=\arg\min_{x,y}\frac{1}{2}||x-(x_k-\gamma \nabla_x f_k)||^2_2+\frac{1}{2}||y-(y_k-\gamma\nabla_y f_k)||^2_2
> $$
> $$
> \qquad \qquad \quad \hbox{s.t.}\quad \nabla_x h_k^\top (x-x_k) + \nabla_y h_k^\top (y-y_k) + \gamma\alpha \rho_k \leq 0,\quad x\in X.
> $$
>
> The main challenge with this reformulated subproblem is that it may fail to admit any feasible solution. To mitigate this issue, one can impose the Linear Independence Constraint Qualification (LICQ), i.e., suppose $X= \\{ x | Ax\leq b \\} $ for some $A = [a_i^\\top]_{i=1}^r \\in \\mathbb{R}^{r \\times n}$ and $b \\in \\mathbb{R}^r$ then
>
> $(a_i)_{i=1}^r, \\nabla_x h(x,y)$
>
> are linearly independent. This condition guarantees the existence of a KKT solution for the original problem and ensures the feasibility of the subproblem.
>
> Second, if the lower-level problem includes a convex constraint set $Y$, the situation becomes more challenging because the presence of $Y$ renders the lower-level solution map non-differentiable. More precisely, using the first-order optimality condition for the lower-level problem, the stationarity-based reformulation of the bilevel problem can be written as $\min_{x,y}~f(x,y)\quad \text{s.t.}\quad 0\in \nabla_y g(x,y)+\mathcal N_Y(y)$. This leads to a nonconvex minimization problem with an inclusion-type constraint.
>
> Following our approach, one might attempt to convert this inclusion into a single equality constraint using a fixed-point characterization: $\min_{x,y}~f(x,y)\quad \text{s.t.}\quad h(x,y)=||y-\mathcal P_Y(y-\gamma \nabla_y g(x,y))||^2$,
> where $\mathcal P_Y$ denotes the projection onto the set $Y$. However, in this setting, the resulting constraint $h$ is *not differentiable*, which makes the overall problem highly challenging. To the best of our knowledge, addressing this issue in full generality (e.g., when $Y$ also depends on $x$) remains an open question that we will study in future work.

---

> ### Author Response · Authors · 2026-02-17
> **Rebuttal of Review 2ChR**
>
> **R4 (cont.)**
>
> In the conclusion section, we have added a new paragraph regarding the extension of the proposed work to constrained bilevel problems. This paragraph is as follows:
>
> ``An important future work is to extend the proposed framework to constrained bilevel optimization. In general, the problem of constrained bilevel optimization has been studied extensively [1-3] and more. The presence of lower-level constraints introduces additional challenges, making the problem significantly harder to solve. For instance, when considering lower-level constraints, the lower-level optimality conditions are no longer given simply by $\nabla_y g(x,y) = 0$; instead, they become a set of KKT conditions.
> This leads to the so-called KKT reformulation of bilevel optimization, which transforms the problem into a mathematical program with complementarity constraints (MPCC). It has been shown that this reformulation is generally not equivalent to the original problem [4], even if the lower-level problem (both objective and constraints) is convex. The reason is that the complementary slackness condition violates standard constraint qualifications (such as LICQ or MFCQ) at every feasible point. Consequently, the first-order necessary condition—that a solution of the bilevel problem must satisfy the KKT conditions—may no longer hold. Due to these challenges, we defer analyses of bilevel optimization problems with lower-level constraints to future work.''
>
> [1] Dempe, S. and Zemkoho, A.B., 2012. On the Karush–Kuhn–Tucker reformulation of the bilevel optimization problem. Nonlinear Analysis: Theory, Methods \& Applications, 75(3), pp.1202-1218.
>
> [2] Khanduri, P., Tsaknakis, I., Zhang, Y., Liu, J., Liu, S., Zhang, J. and Hong, M., 2023, July. Linearly constrained bilevel optimization: A smoothed implicit gradient approach. In International Conference on Machine Learning (pp. 16291-16325). PMLR.
>
> [3] Dempe, S. and Mehlitz, P., 2025. Duality-based single-level reformulations of bilevel optimization problems. Journal of Optimization Theory and Applications, 205(2), p.26.
>
> [4] Dempe, S. and Dutta, J., 2012. Is bilevel programming a special case of a mathematical program with complementarity constraints?. Mathematical programming, 131(1), pp.37-48.

---

> ### Author Response · Authors · 2026-02-17
> **Rebuttal of Review 2ChR**
>
> **R5.**
> We thank the reviewer for this valuable observation. Unfortunately, we did not observe any improvement in the convergence rate when the lower-level problem is purely convex. The main reason is that, under the stationary-seeking reformulation employed in our work, the resulting constraint $\nabla_y g(x,y)=0$ can still be highly nonlinear and therefore nonconvex. Consequently, establishing feasibility without additional assumptions is still challenging, even when the lower-level objective is convex. This motivated us to introduce the regularity condition, which promotes feasibility when the gradient of the constraint vanishes. Nonetheless, we would like to emphasize that the equivalence between the proposed reformulation (3) and the original bilevel problem (BLO) holds when the lower-level function $ g(x, \cdot) $ is convex or satisfies the (weak) PL condition (Xiao et al., 2023a) for any $x$, i.e.,
> $\nabla_y g(x,y^*(x))=0$
>
> if and only if $y^*(x)\in\arg\min_{y}g(x,y)$.
>
> **R6.**
> Thank you for your question. The function $\rho(x,y)$ in the QP serves as a "barrier-like" term that promotes the reduction of infeasibility. More specifically, the constraint in the QP is designed to select a  *descent direction* $\Delta^z=(\Delta^x,\Delta^y)$ for the constraint function $h$, i.e., $\nabla h(z_k)^\top \Delta^z<0$. The term $\alpha \rho_k$ controls the magnitude of this descent, and hence the convergence rate.
> The choice of $\rho_k$ was deliberately made for algebraic reasons to facilitate the proof.
> However, here we also provide a control-theoretic intuition and will include it in the revised manuscript if needed.
>
> Denote $z = (x^\top, y^\top)^\top$. To enforce $h(z)$ to converge to zero asymptotically, a sufficient condition, which is more general than exponential convergence, is that $h(z)$ satisfies the following condition:
> $$
> \dot{h}(z) + \alpha (h(z)) \leq 0
> $$
> Here $\alpha$ is a class-$\mathcal{K}$ function, i.e., a function $\alpha:(-b, a) \mapsto \mathbb{R}$ for some $a, b > 0$ that is strictly increasing and satisfies $\alpha(0) = 0$.
> The above inequality ensures that $h$ will converge asymptotically, and the choice of $\alpha$ determines the rate of convergence. For example, choosing $\alpha$ to be a linear function results in exponential convergence:
> $$
> \dot{h}(z(t)) + \gamma h(z(t)) \leq 0 \implies h(t) \leq h(0) e^{-\gamma t} \quad \gamma >0.
> $$
>
> As another example, if we instead use the square root function, we obtain a finite convergence rate:
> $$
> \dot{h}(z(t)) + \gamma h(z(t))^{1/2} \leq 0 \implies
> h(t) \leq (\sqrt{h(0)} - \frac{\gamma}{2} t)^2 \quad \text{for } 0 \leq t \leq \frac{2\sqrt{h(0)}}{\gamma}
> $$
> In our context, $\rho_k$ plays the role of the $\alpha$-function. Specifically, under a Polyak--Lojasiewicz (PL) condition on $h$, i.e., $\\|\nabla h(z)\\|^2 \geq 2\mu h(z)$ for some $\mu>0$ (noting that $h(z^\star) = 0$), we consider the following cases:
>
> 1. $\\rho_k = \\|\\nabla h(z)\\|^2$:
>    In this case, by the PL assumption, we recover the exponential convergence rate discussed above:
>
>    $$
>    \\dot{h}(z) + 2\\alpha \\mu h(z)
>    \\leq
>    \\dot{h}(z) + \\alpha \\rho_k
>    \\leq 0.
>    $$
>
> 2. $\\rho_k = \\|\\nabla h(z)\\| \\sqrt{h(z(0))}$:
>    In this case, again using the PL assumption, we recover the finite-time convergence rate discussed above:
>
>    $$
>    \\dot{h}(z) + \\alpha \\sqrt{2\\mu h(z(0)) h(z)}
>    \\leq
>    \\dot{h}(z) + \\alpha \\rho_k
>    \\leq 0.
>    $$
>
>
> In this context, our algorithm can be viewed as a discretization of the above condition.
>
> **R7.**
> The stationarity measure introduced in Definition 2.4 of our paper is indeed the $\epsilon$-KKT measure of problem (3). This quantity is a standard metric used in constrained optimization and is also widely adopted in bilevel optimization problems, particularly when the lower-level problem is not strongly convex. For example, in (Lie et al. 2022), by employing a value-function reformulation, the authors establish the convergence rate of their method directly in terms of the KKT conditions of the reformulated problem, which has been shown to imply a KKT point of problem (3) under constant rank constraint quantification (CRCQ). The $\epsilon$-KKT metric of problem (3) is also used in (Xiao et al., 2023a). In (Chen et al., 2024) and (Huang, 2024), the authors adopt an $\epsilon$-implicit gradient metric $\|\nabla \ell(x)\|^2\leq \epsilon$ where $\ell(x)=\min_{x,y\in Y^*(x)} f(x,y)$. Note that under the PL condition and uniqueness of the lower-level solution, these metrics are equivalent -- see (Xiao et al., 2023a) and (Huang, 2024).

---

### Review · Reviewer_hR3U · 2025-11-28

**Summary Of Contributions:**

The paper considers bilevel optimization problems where the inner problem is not necessarily strongly convex. Building on the Safe Gradient Flow ODE for bilevel optimization proposed by Sharifi et al. (2025), the authors propose a discretization of this ODE to reach a KKT point of a reformulated problem in which inner optimality is replaced by inner stationarity. The authors analyze two variants of their algorithm depending on the choice of the function $\rho$. When $\rho(x,y) = \lVert\nabla h(x,y)\rVert^2$, the algorithm is shown to achieve $\mathcal{O}(\epsilon^{-3/2})$ complexity, while, in the case where $\rho(x,y) = \lVert\nabla h(x,y)\rVert\sqrt{h(x_0, y_0)}$, the complexity becomes $\mathcal{O}(\epsilon^{-3})$. The algorithms are evaluated numerically on a data hypercleaning task and on synthetic problems.

**Audience:**

Yes

**Audience Explanation:**

Bilevel optimization is an important field that have attracted lots of attention in the ML community due to its wide range of applications, *e.g.* hyperparameter selection or neural architecture search. Therefore, this submission is of interest of some of the TMLR audience.

**Claims And Evidence:**

No

**Claims Explanation:**

### Theory
* **T1**: The theoretical analysis is sound
* **T2**: The paper does not require PL assumption for the inner problem.

### Experiments
The theoretical part is solid, but I have several concerns regarding the experiments:

* **E1**: Some experimental details are missing, making them not reproducible:
    * *Section 5.1.*: the sizes of the validation and training set are not specified.
    * *Appendix C.1*: the generating process of the different synthetic problems is not sufficiently detailed.
* **E2**:  It is unclear why some competitor algorithms are compared to Algorithm 1 only in a subset of the experiments.
    * AIDBiO is excluded from the neural network classifier experiment due to “he high computational burden of calculating the Hessians of neural networks”. However, AIDBiO does not require explicit Hessian computation—only Hessian–vector products—which can be efficiently computed via automatic differentiation, as noted by the authors themselves (Remark 3.1) and explained in [1,2].
    * BOME is omitted from the high-dimensional DHC experiment without explanation.
    * VPBGD appears in the experiment of Section 5.1.1 but not in the experiment of Section 5.1.
* **E3**: The numerical performances are displayed in function of iterations.However, different algorithms have different per-iteration complexities. For instance, AIDBiO and BOME are two-loop algorithms, while the proposed algorithm is single-loop. Moreover, the proposed algorithm uses second-order information whereas BOME does not. Therefore, iteration-based plots do not yield a fair comparison of the practical performances of the algorithms.
* **E4**: In figure 2, only one variant of the algorithm is displayed, and we don't know which one and why only one. In figure 3, only the variant of Theorem 4.2 is displayed.
* **E5**: I do not understand the purpose of Figure 4, particularly the two left-most panels. What does the paramete $K$ in the legend represent? If it is a number of iterations, why are the curves are not superposed since the x-axis is already iterations?


[1] B. A. Pearlmutter. *Fast Exact Multiplication by the Hessian*. Neural Computation, 1994.

[2] M. Dagréou, P. Ablin, S. Vaiter, T. Moreau. *How to compute Hessian-vector products?*. ICLR blogpost, 2024.

**Requested Changes:**

Changes **C1–C3** would improve the paper but are not required for acceptance.

Changes **C7–C9** are necessary for the experimental section to be consistent.

### Clarity
* **C1**: In Table 1, specify the convergence criterion used in each paper. Since each paper uses different criteria, the complexity results are not directly comparable and the current presentation may mislead readers.
* **C2**: Explain the motivation for using ODE (5) to solve (4). This would make the paper easier to follow.
* **C3**: Clarify the difference between the proposed method and the Euler discretization of (5), including pros and cons. This is not sufficiently clear in the current version.

### Minor
* **C4**: Proof of lemma 4.1: "$C^g_y$" -> "$C_g$"
* **C5**: Equation (9), second line: $\nabla_z f(z_k)$ -> $\nabla f(z_k)$
* **C6**: In Figures 3-5, in the legend, "Theorem 4.1" should be "Theorem 4.2"?

### Critical changes
* **C7**: Describe the experimental settings in sufficient detail to make the experiments reproducible and the results interpretable.
* **C8**: Present results in a fairer manner, e.g., in terms of wall-clock time instead of iterations.
* **C9**: Either include all missing solvers in all experiments or provide clear justification for their absence.

---

> ### Author Response · Authors · 2026-02-25
> **Rebuttal of Review hR3U**
>
> ***Experiments***
>
> **RE1.**
> For the DHC experiments, we use 5000, 2500, and 2500 samples for $D_{tr}$, $D_{val}$, and $D_{test}$, respectively.
> For the synthetic examples in Sections C.1 and C.3, following the setup of Sharifi et al. (2025), to generate the random matrix $H$, we first sample its eigenvalues uniformly between 0.1 and 1 (ensuring the condition number is at most 10), and then apply a QR decomposition on a random matrix to construct the final matrix using the generated eigenvalues. The remaining vectors are generated randomly.
> In Section C.2, we follow the exact setup outlined by Liu et al. (2022).
> We have added additional details to each section to improve clarity.
> Moreover, in the interest of transparency, the code will be publicly released after the review process. Due to the double-blind nature of the review, we can provide an anonymous link at this stage if the reviewer deems it necessary.
>
>
> **RE2.**
> - AIDBiO requires running multiple iterations of the conjugate gradient method, during which the Hessian $ \nabla_{yy}^2 g(x,y) $ remains fixed since neither $x$ nor $y$ is updated.
> Specifically, AIDBiO aims to compute the hypergradient in equation (1a) by estimating the auxiliary variable $v(x,y)$ via K steps of the conjugate gradient method applied to (1b).
> In essence, the method repeatedly computes Hessian--vector products of the form
> $$
> \nabla_{yy}^2 g(x,y) v_i, \qquad i \in \{0,\dots,K\}.
> $$
> As a result, there are two possible implementation strategies.
> The first is to explicitly compute the Hessian $ \nabla_{yy}^2 g(x,y) $ and perform matrix--vector multiplications within the conjugate gradient algorithm, which is more suited for low-dimensional problems.
> The second is to apply PyTorch automatic differentiation $K$ times, once per conjugate gradient iteration, which was implemented in the codebase of (Ji et al., 2021).
> Both of these implementations cause an overhead compared to the rest of the methods we compare with in the paper.
> In addition to this issue, another limitation is that AIDBio was designed for problems where the lower-level (LL) problem is strongly convex, i.e., $v(x,y)$ is uniquely defined. For these reasons, we only compare with AIDBio in the small-scale strongly convex setting and exclude it from the large-scale experiments.
> We have also added these arguments and updated the claims in the paper (see section 5 of the revised manuscript).
>
> - We have added BOME and VPBGD to all experiments in Section~5 and the Appendix.
> We performed a grid search to identify an approximately optimal set of hyperparameters for each experimental setting.
> It can be seen that the proposed method consistently outperforms both BOME and VPBGD across the majority of settings, while achieving comparable performance in the remaining cases.
>
>
> **RE3.**
> The reviewer is correct to raise this concern. Indeed, this was one of the challenges we faced when designing the experimental setup. In our experiments, we compare methods ranging from single-loop to double-loop, and from approaches that use only first-order information to those that also exploit second-order information.
> To have a fair comparison, we conducted experiments in which methods were given a **fixed time budget** instead of a fixed number of iterations, and we report these results in the appendix (see Section C.2). It can be observed that, except for AIDBio—which incurs a higher computational cost, as explained above—the remaining algorithms exhibit comparable runtimes. We have presented the time-budget results in the appendix for completeness.
>
> **RE4.**
> We have now included both variants in both Figures 2 and 3.
>
> **RE5.**
> This point is slightly nuanced. If one examines the algorithm closely, the step size is a function of the number of iterations $K$. Consequently, running the algorithm for a longer duration reduces the step size. This explains why the figures are not superposed. However, the smaller step size allows the method to converge to a better local solution. This experiment was designed to show this trade-off.

---

> > ### Author Response · Authors · 2026-02-25
> > **Rebuttal of Review hR3U**
> >
> > ***Requested Changes***
> >
> > **R1.**
> > Thank you for your thoughtful comment. In the revised manuscript, we added the convergence criterion in section 1.2 when discussing each paper's result. In particular, in our paper, we use an $ \epsilon $-KKT metric of problem (3) given in Definition 2.4. This quantity is a standard metric used in constrained optimization and is also widely adopted in bilevel optimization problems, particularly when the lower-level problem is not strongly convex. For example, in (Lie et al. 2022), by employing a value-function reformulation, the authors establish the convergence rate of their method directly in terms of the KKT conditions of the reformulated problem, which has been shown to imply a KKT point of problem (3) under constant rank constraint quantification (CRCQ). The $\epsilon$-KKT metric of problem (3) is also used in (Xiao et al., 2023a). In (Chen et al., 2024) and (Huang, 2024), the authors adopt an $\epsilon$-implicit gradient metric $\|\nabla \ell(x)\|^2\leq \epsilon$ where $\ell(x)=\min_{x,y\in Y^*(x)} f(x,y)$. Note that under the PL condition and uniqueness of the lower-level solution, these metrics are equivalent -- see (Xiao et al., 2023a) and (Huang, 2024).
> >
> > **R2.**
> > In the revised manuscript, we have expanded the explanation of the convex QP in Eq.~(5) to clarify its role and intuition. Specifically, we now describe how the QP selects velocity vectors $(\dot{x}, \dot{y})$ that remain as close as possible to the nominal gradient-descent direction on the upper-level objective while satisfying the linear constraint $\dot{h} + \alpha h = 0$. We further explain that this constraint admits the solution $h(t) = h(0)e^{-\alpha t}$, thereby ensuring exponential decay of $h$ toward satisfying the lower-level condition. This additional explanation appears right after Eq. (5).
> >
> > **R3.**
> > The constraint of the QP in Eq.(5) guarantees exponential convergence of \(h\) to zero, in continuous time, through the linear term $\alpha h$. However, an explicit Euler discretization of the continuous-time dynamics defined by (5) does not, in general, preserve this exponential convergence unless restrictive conditions are imposed on the lower-level objective. To address this limitation, we replace the linear term \(\alpha h\) in(5) with a nonlinear term $\alpha \rho_k$ to obtain the new QP in (7). We then apply Euler discretization on the continuous-time dynamics defined by the modified QP in (7). The nonlinear function $ \rho_k $ is carefully designed to ensure infeasibility decay under less restrictive assumptions, as shown in Corollaries 4.3 and 4.6.
> >
> > **R4-R5.**
> > The typos have been addressed.
> >
> > **R6.**
> > Thank you for pointing this out. To avoid confusion, we have changed the legends, where now the first algorithm is denoted by $\rho = ||\nabla h(x,y)||^2$ and the second algorithm by $\rho = ||\nabla h(x,y)||\sqrt{h(x,y)}$.
> >
> > **R7.**
> > We have added an additional explanation to the revised manuscript. We also refer the reviewer to the RE1 discussion above.
> >
> > **R8.**
> > We have added a section to the appendix to address this concern. We refer the reviewer to the RE3 discussion above.
> >
> > **R9.**
> > We have re-run all experiments to address this comment. We refer the reviewer to the discussions in RE2 and RE4 above.

---

> > > ### Comment · Reviewer_hR3U · 2026-02-27
> > >
> > > Dear authors,
> > >
> > > Thank you for the response. Could you please upload the revised manuscript?
> > > For the code, it is also possible to submit it as supplementary material.
> > >
> > > Best

---

> > > > ### Author Response · Authors · 2026-03-25
> > > > **Replying to Comment by Reviewer hR3U**
> > > >
> > > > Dear Reviewer,
> > > >
> > > > The revised manuscript and the code (in the supplementary material) have now been submitted.
> > > >
> > > > Best,
> > > > Authors

---

### Review · Reviewer_6YW6 · 2026-03-12

**Summary Of Contributions:**

This paper studies nonconvex bilevel optimization and proposes a perturbed gradient descent method that incorporates a quadratic program (QP) correction step to improve satisfaction of lower-level optimality conditions during optimization. At each iteration, the algorithm computes a gradient-based update and solves a QP that adjusts the search direction to reduce violations of lower-level stationarity. The authors provide convergence guarantees showing that the method achieves approximate stationarity under certain regularity conditions. Empirical results are reported on several standard bilevel learning benchmarks, including data hyper-cleaning and coreset selection, comparing the proposed method to existing bilevel optimization approaches.

## Strengths
- **Challenging bilevel setting**: The paper studies the nonconvex--nonconvex bilevel optimization setting, which is more general than the commonly studied convex case. Providing convergence guarantees in this setting is technically challenging and represents a meaningful theoretical direction.
- **Clear algorithmic structure**: The proposed method combines gradient descent with a QP-based correction step designed to reduce violations of lower-level optimality conditions. The algorithm is conceptually simple and the presentation of the method is clear.
- **Evaluation on standard bilevel benchmarks**: The experimental evaluation considers commonly used bilevel learning tasks such as data hyper-cleaning and coreset selection and compares against several established baselines from the bilevel optimization literature.

## Weaknesses
- **Hidden computational cost in experiments**:
The experimental results report performance primarily in terms of iteration counts. However, the proposed algorithm requires second-order oracle calls at each iteration, while several baselines (e.g., BOME) rely solely on first-order gradient updates. Although the authors note in Remark 3.1 that these products can be computed via automatic differentiation without forming the full Hessian, they still are likely much more expensive than simple gradient evaluations. As a result, I expect that the per-iteration cost differs significantly across methods, and iteration-count comparisons alone may overstate the practical advantage of the proposed approach. Reporting wall-clock runtime would provide a fairer comparison.
- **Experimental robustness**: Results are presented as single optimization trajectories (validation loss and test accuracy versus iterations), without reporting averages or variability across multiple runs. Since performance depends on random initialization, data splits, and label corruption patterns, it is difficult to assess whether the observed differences between methods are statistically meaningful. This is particularly important given that some comparisons (e.g., Figure 2, low-dimension DHC) show relatively modest gaps between methods. Reporting means and confidence intervals across several random seeds would considerably strengthen the empirical conclusions.
- **Baseline consistency across experiments**:
Different experiments include different subsets of baselines, and the rationale is not always clearly justified. For example, AIDBiO is excluded from the neural network experiment due to the computational cost of Hessian computations, yet the proposed method also relies on second-order information, and no discussion is provided on why its own overhead remains acceptable in that setting. More broadly, several methods featured in the comparison table (Table 1), e.g., GALET and HJFBiO, are never evaluated against experimentally, despite being the most directly relevant competitors in the nonconvex-PL setting. Clarifying why specific baselines were included/excluded and, where possible, adding comparisons to these methods would make the empirical evaluation more convincing.

**Audience:**

Yes

**Audience Explanation:**

Yes, continuous nonconvex bilevel optimization is a relatively active area in ML research and it will be of interested to the general ML community.

**Broader Impact Concerns:**

None.

**Claims And Evidence:**

No

**Claims Explanation:**

Most claims are well supported, but I do think more empirical evaluation is needed to really access the practical advantages this approach may have.

**Requested Changes:**

Weaknesses 1-3 need to be addressed and justified.  I'd also appreciate a more transparent/summarized discussion of any limitations before the conclusion.

---

> ### Author Response · Authors · 2026-03-25
> **Rebuttal of Review 6YW6**
>
> **R1.** Thank you for this important comment. We agree that comparing time complexity provides a clearer comparison between the considered methods.
> For this reason, in the revised manuscript we conducted experiments in which methods were given a **fixed time budget** instead of a fixed number of iterations, and we report these results in the appendix (see Section C.2). It can be observed that, except for AIDBio—which is noticeably more expensive due to its repeated conjugate-gradient/Hessian-vector-product computations—the remaining methods exhibit comparable runtimes. We have presented the time-budget results in the appendix for completeness.
>
>
> **R2.** We thank the reviewer for this insightful comment. We agree with the comment, and in response, we have updated the experimental section of the paper to report results aggregated over multiple runs. Specifically, all plots now show the mean performance along with confidence intervals computed over 5 independent random seed initializations. We believe this provides a more reliable and informative comparison between methods and strengthens the empirical conclusions of the paper.
>
> **R3.** We thank the reviewer for this comment. We have now added BOME and VPBGD to all experiments in Section 5 and the Appendix.
> Regarding the comparison with AIDBiO, it requires running multiple iterations of the conjugate gradient method, during which the Hessian $\nabla_{yy}^2 g(x,y)$ remains fixed since neither \(x\) nor \(y\) is updated.
>
> Specifically, AIDBiO aims to compute the hypergradient in equation (1a) by estimating the auxiliary variable \(v(x,y)\) via \(K\) steps of the conjugate gradient method applied to (1b).
> In essence, the method repeatedly computes Hessian--vector products of the form
> $$
> \nabla_{yy}^2 g(x,y)\, v_i, \qquad i \in \{0,\dots,K\}.
> $$
> As a result, there are two possible implementation strategies.
> The first is to explicitly compute the Hessian $ \nabla_{yy}^2 g(x,y) $ and perform matrix--vector multiplications within the conjugate gradient algorithm, which is more suited for low-dimensional problems.
> The second is to apply PyTorch automatic differentiation $K$ times, once per conjugate gradient iteration, which was implemented in the codebase of (Ji et al., 2021).
> Both of these implementations cause an overhead compared to the rest of the methods we compare with in the paper.
> In addition to this issue, another limitation is that AIDBio was designed for problems where the lower-level (LL) problem is strongly convex, i.e., $v(x,y)$ is uniquely defined. For these reasons, we only compare with AIDBio in the small-scale strongly convex setting and exclude it from the large-scale experiments.
> We have also added these arguments and updated the claims in the paper (see section 5 of the revised manuscript).
>
> We agree that HJFBiO is theoretically relevant to the nonconvex-PL literature. However, our empirical section was designed primarily to evaluate the practical effectiveness of the proposed method in large-scale settings against scalable baselines, which is why we prioritized BOME and VPBGD. In contrast, HJFBiO is analyzed under an additional local nondegeneracy assumption on the lower-level problem, namely that the lower-level Hessian at $y^*(x)$ has spectrum bounded away from zero. Its hypergradient estimator also relies on a clipping/projection of the lower-level Hessian spectrum. In the large-scale neural-network settings considered in our experiments, such structural conditions are difficult to verify in practice. Moreover, as discussed in Remark 1.1, GALET requires an additional regularity condition that narrows the class of problems to those with a strongly convex lower-level objective function. Therefore, we do not view HJFBiO and GALET as the most informative practical baseline for the experimental question studied here.

---

> ### Author Response · Authors · 2026-03-25
> **Rebuttal of Review 6YW6**
>
> **R4.** We have added section called ''Limitations \& Future Works'' before the conclusion section:
>
> ''
> Despite strong theoretical and empirical results, the proposed method has some limitations. A primary issue is the cost of Hessian-related operations needed to enforce the lower-level optimality condition. In particular, our method relies on Hessian-vector products (see Remark 3.1), which, while efficiently implemented via automatic differentiation, can still become expensive in high-dimensional settings, limiting scalability compared to fully first-order methods. Developing inexpensive approximations remains an important direction for future work.
> Another important future direction is to extend the proposed framework to constrained bilevel optimization. In general, the problem of constrained bilevel optimization has been studied extensively [1-3] and more. The presence of lower-level constraints introduces additional challenges, making the problem significantly harder to solve. For instance, when considering lower-level constraints, the lower-level optimality conditions are no longer given simply by $\nabla_y g(x,y) = 0$; instead, they become a set of KKT conditions.
> This leads to the so-called KKT reformulation of bilevel optimization, which transforms the problem into a mathematical program with complementarity constraints (MPCC). It has been shown that this reformulation is generally not equivalent to the original problem [4], even if the lower-level problem (both objective and constraints) is convex. The reason is that the complementary slackness condition violates standard constraint qualifications (such as LICQ or MFCQ) at every feasible point. Consequently, the first-order necessary condition—that a solution of the bilevel problem must satisfy the KKT conditions—may no longer hold. Due to these challenges, we defer analyses of bilevel optimization problems with lower-level constraints to future work.
> ''
>
> [1] Dempe, S. and Zemkoho, A.B., 2012. On the Karush–Kuhn–Tucker reformulation of the bilevel optimization problem. Nonlinear Analysis: Theory, Methods \& Applications, 75(3), pp.1202-1218.
>
> [2] Khanduri, P., Tsaknakis, I., Zhang, Y., Liu, J., Liu, S., Zhang, J. and Hong, M., 2023, July. Linearly constrained bilevel optimization: A smoothed implicit gradient approach. In International Conference on Machine Learning (pp. 16291-16325). PMLR.
>
> [3] Dempe, S. and Mehlitz, P., 2025. Duality-based single-level reformulations of bilevel optimization problems. Journal of Optimization Theory and Applications, 205(2), p.26.
>
> [4] Dempe, S. and Dutta, J., 2012. Is bilevel programming a special case of a mathematical program with complementarity constraints?. Mathematical programming, 131(1), pp.37-48.

---

### Decision · Action_Editor_CMaa · 2026-04-17

**Recommendation:** Accept as is

**Audience:**

Yes

**Audience Explanation:**

Bilevel optimization is an active area in machine learning in the last years, with applications in hyperparam tuning, meta-learning, and adversarial learning that will interest a portion of the TMLR community.

**Claims And Evidence:**

Yes

**Claims Explanation:**

The authors provide rigorous convergence analysis since the start and have strengthened the empirical evaluation during the revision, addressing reviewer concerns about reproducibility and timing fairness.